# Simulation-aided characterization of a versatile water condensation particle counter for atmospheric airborne research

Fan Mei[1], Steven Spielman[2], Susanne Hering[2], Jian Wang[3], Mikhail Pekour[1], Gregory Lewis[2], Beat Schmid[1], Jason Tomlinson[1], Maynard Havlicek[4]

[1]Pacific Northwest National Laboratory, Richland, WA, 99352, USA
[2]Aerosol Dynamics Inc., Berkeley, CA, 94710, USA
[3]Washington University in St. Louis, St. Louis, MO, 63130, USA
[4]TSI Incorporated, Shoreview, MN, 55126, USA

*Correspondence to*: Fan Mei (fan.mei@pnnl.gov)

**Abstract**

Capturing the vertical profiles and horizontal variations of atmospheric aerosols often requires accurate airborne measurements. With the advantage of avoiding health and safety concerns related to the use of butanol or other chemicals, water-based condensation particle counters have emerged to provide measurements under various environments. However, airborne deployments are relatively rare due to the lack of instrument characterization under reduced pressure at flight altitudes. This study investigates the performance of a commercial "versatile" water CPC (vWCPC, Model 3789, TSI, Shoreview, MN, USA) under various ambient pressure conditions (500 – 920 hPa) with a wide range of particle total number concentrations (1,500 ~ 70,000 cm$^{-3}$). The effect of conditioner temperature on vWCPC 3789 performance at low pressure is examined through numerical simulation and laboratory experiments. We show that the default instrument temperature setting of 30°C for the conditioner is not suitable for airborne measurement and that the optimal conditioner temperature for low-pressure operation is 27 ℃. Under the optimal conditioner temperature (27 ℃), the 7 nm cut-off size is also maintained. Additionally, we show that insufficient droplet growth becomes more significant under the low-pressure operation. The counting efficiency of the vWCPC 3789 can vary up to 20% for particles of different chemical compositions (e.g., ammonium sulfate and sucrose particles). However, such variation is independent of pressure.

## 1 Introduction

Atmospheric aerosol particles are one of the key components of the atmosphere. The currently known ambient aerosol has a size range over several magnitudes and consists of complex chemical compositions, which vary with size, origin, age, and atmospheric processing. This tiny but complicated particulate matter plays a remarkable role in climate change (Seinfeld et al., 2016) and human health (Anderson et al., 2020; Lighty et al., 2000; Pöschl, 2005). To understand the variation of atmospheric aerosol and its production, distribution, and evolution paths, Friedlander introduced a conceptual framework for characterizing instruments used for aerosol measurements (Friedlander, 1970, 1971). Following his framework, the size distribution and number concentration of atmospheric aerosol particles are detected through electrostatic methods and condensational growth. The latter approach is the only technique available for detecting uncharged sub-50 nm particles. Consequently, it has become the dominant technique for assessing the integrated concentration of particles larger than a minimum size.

Since P. J. Coulier and J. Aitken published their observations dealing with the role of a fine airborne particle in the vapor condensation process in 1875 and 1880 separately (Spurny, 2000), the need to understand the phenomena has inspired the development of several particle counting instruments and led to various methods to quantify their performance under different operating conditions (Kangasluoma and Attoui, 2019; McMurry, 2000a; McMurry, 2000b). Several reviews have discussed the

development of this technique in atmospheric aerosol measurements (Curtius, 2006; Kerminen et al., 2018; Kulmala et al., 2004; McMurry, 2000b). McMurry divided the history of condensation nucleus counters into two main sections through the end of the twentieth century – the development of expansion-type instruments and steady-flow condensation nucleus counters (McMurry, 2000a). Sem describes the designs of three commercial condensation particle counters (CPCs) and characterizes the particle diameter with 50% detection efficiencies of a TSI CPC (3025A, 3022A, and 3010) (Sem, 2002). Two recent comprehensive reviews by Kangasluoma et al. focus on developing instruments that can measure the particle size distribution down to the size of large molecules (Kangasluoma and Attoui, 2019; Kangasluoma et al., 2020). Kangasluoma et al. provide an in-depth review of the effort to advance the technology toward sub-10 nm size distribution measurements, summarize the current understanding of the characteristics of several systems, and identify instrumental limitations and potential advances for accuracy improvement in sub-3 nm particle counting.

In general, an airborne CPC operates by the same principle as standard ground-based CPCs discussed above. It is essential to allow sufficient supersaturation generated inside the condenser for a continuous flow CPC as discussed in this manuscript. Previous studies have described the required modifications of a commercial CPC for aircraft operation and demonstrated how to characterize such sensors under low operating pressures down to 150-200 hPa (Hermann et al., 2005; Hermann and Wiedensohler, 2001; Schröder and Ström, 1997). In the effort to capture the rapid change of the particle concentration from the boundary layer to the stratosphere, the nucleation mode aerosol size spectrometer (NMASS) has been used on research aircraft since 1999. A comprehensive description of the NMASS, its uncertainties under laboratory studies and operation during the Atmospheric Tomography (ATom) mission was published in 2018. Two NMASS were comprised of ten parallel CPCs (five for each NMASS) operating at an internal pressure of 120 hPa. They measured the size distribution between 3 and 60 nm and provided a robust analytical foundation to probe the new particle formation event globally (Brock et al., 2019; Williamson et al., 2018).

Although most CPCs use butanol vapor to grow the aerosol particles, researchers notice that the working fluid plays a critical role in determining the size detection limits, as the vapor properties affect how the vapor condenses upon the particle to enlarge its size for detection. Stolzenburg and McMurry first described the effect of working fluid on size-dependent activation efficiencies with the laminar flow ultrafine condensation particle counter (Stolzenburg and McMurry, 1991), then theoretically described the effect of the working fluid. Experimental studies by Iida et al. (Iida et al., 2009) complemented their theory. Magnusson et al. concluded that working fluids with high surface tension (such as water and glycerol) could lead to a smaller activation size and reduce the lower size detection limit in the CPCs (Magnusson et al., 2003b). For airborne CPCs, 1-butanol and FC-43 (Hermann et al., 2005)are used as a working fluid, and FC-43 shows better performance below 200 hPa. As a working fluid, water avoids the health and safety concerns of butanol or other chemicals. Over the last few decades, mixing type water-based condensation systems have been used to capture particles for online chemical speciation instruments (Khlystov et al., 1995) and high-flow condensation particle counting (Parsons and Mavliev, 2001).

Additionally, water-based CPCs reduce the requirement of chemical storage, maintenance effort, and ventilation system and eliminate water condensation and absorption into alcohol working fluids during operation in humid environments (Liu et al., 2006). However, due to a three times higher mass diffusivity of water compared to butanol, it is very challenging to use water as the condensing fluid in the above thermally, diffusive laminar flow CPC. Therefore, the measurement principle has to be changed (Hering et al., 2005). Hering and Stolzenburg (2005) introduced the concept of a warm, wet-walled condenser for water-based condensational growth. The first implementation of this concept was the two-stage water CPC (Hering et al., 2005). The performance of several versions of this two-stage water-based CPC was intensively evaluated in the 21st century (Biswas et al., 2005; Hakala et al., 2013; Hering et al., 2005; Iida et al., 2008; Keller et al., 2013; Kupc et al., 2013; Kurten et al., 2005; Liu et al., 2006; Mordas et al., 2008; Petaja et al., 2006). In 2014, Hering and co-workers further improved the laminar flow water CPC with

a third stage to moderate the temperature profile between the growth tube and optics (Hering et al., 2014). This advanced design enables the capture of water vapor by the third moderator stage, such that conditions inside the instrument can be self-sustaining with regard to water consumption under moderate to high relative humidity (Hering et al., 2019; Kangasluoma and Attoui, 2019)). This feature has been commercialized in the "MAGIC" CPC (Moderated Aerosol Growth with Internal water Cycling, ADI). This concept also enables a higher temperature for the second 'initiator' stage, providing supersaturation can be created between the conditioner and initiator, and leads to the activation of 1 nm particles without homogeneous water droplet formation (Hering et al., 2017; Kangasluoma and Attoui, 2019). It further allows flexibility in operating temperatures and a lower detection threshold, and hence has been named the "versatile" water CPC, or vWCPC. The vWCPC has been commercialized by TSI as the Model 3789. Many field deployments confirm that the water-based CPC has comparable performance to a butanol-based CPC in terms of cut-off size and detection efficiency when examining urban pollution and diesel combustion aerosol (Franklin et al., 2010; Jeong and Evans, 2009; Kaminsky et al., 2009; Keller et al., 2013; Lee et al., 2013; Sharma et al., 2011; Tsang et al., 2008). Based on these promising research results, it is desirable to explore the advanced water-based CPC for airborne measurements.

Airborne aerosol measurements provide researchers with *in situ* atmospheric properties across various spatial scales up to thousands of kilometers. However, it also creates design and characterization challenges due to the rapid change in environmental conditions. The pressure dependency of the counting efficiency of non-water-based CPCs has been explored in several studies, and lower cut-off diameter of the CPCs usually increased with the decrease of the operating pressure (Hermann et al., 2005; Hermann and Wiedensohler, 2001; Seifert et al., 2004; Weigel et al., 2009). However, our literature research shows no records about the water-based CPC being evaluated under low-pressure conditions. Thus, this study characterizes a vWCPC 3789 and its counting efficiency changes under such low-pressure conditions. The current simulation was highly nonlinear for the pressures lower than 500 hPa, and the returned solution failed to converge. In addition, we observed inconsistent behavior in one of three vWCPC 3789 we tested. Thus, this manuscript focuses on the measurements and modeling that were done over the pressure range from 500 hPa to 1000 hPa. This pressure range also spans an altitude of 6000 m (~20,000 ft), which is the upper limit for most drone and balloon operations due to FAA restrictions. We examined the 7-nm configuration rather than the 2-nm configuration because the equilibrium water vapor pressure at the maximum temperature in the 2-nm configuration exceeds 500 hPa. A three-stage operating temperature profile simulation was carried out to understand the supersaturation profile inside the water-based condensational growth tube and guide the optimization of the operation setting. A previously developed particle growth model was used to evaluate the pressure change effect on aerosol particle activation and droplet growth. A simplified theoretical analysis is presented to evaluate effects associated with high particle concentrations. Data at low pressures were obtained using a mono-dispersed aerosol of various chemical compositions compared to a CPC operated at atmospheric pressure and a parallel aerosol electrometer at low pressure.

## 2 Materials and Methods

### 2.1 Instrument description and modification

The vWCPC 3789 tested in this study uses a three-stage growth tube, as described by Hering et al. (2017). A single tube with a 6.3mm ID is lined with a wet, porous wick. It has three temperature regions, referred to as the conditioner, initiator, and moderator, with lengths of 73 mm, 30 mm, and 73 mm, respectively. The aerosol flow is 0.3 L/min. The vWCPC 3789 operates in single-particle count mode up to $2\times10^5$ cm$^{-3}$. The manufacturer provides two default cut-off diameter settings: 2 nm and 7 nm based on the characteristics by Kangasluoma et al. (Kangasluoma et al., 2017), using particles from a heated tungsten wire in nitrogen flow.

For the 7-nm configuration tested here, the factory default temperature settings of the walls of the conditioner, initiator, and moderator regions are: Tcond=30°C, Tini=59°C, and Tmod=10°C.

Several modifications were made to the unit in this study because commercially available vWCPC 3789 are not designed for low-pressure applications, as shown in Fig. 1. First, the testing vWCPC 3789 was tested to ensure it is vacuum-tight. Therefore, the make-up flow port and exhaust port were blocked during the vacuum-tight check. In addition, the water fill bottle was connected

during the vacuum-tight test and low-pressure operations. This step guarantees the instrument operates appropriately under conditions of a significant positive difference between ambient and internal pressures, which mimic the characteristic of the high-altitude operation on an aircraft with a pressurized cabin. Secondly, the vWCPC 3789 monitors the inlet pressure, orifice pressure and nozzle pressure during the operation. Thus, we connected the ambient pressure port and inlet pressure port to the low-pressure manifold, which prevented triggering the warning and error indicator. Thirdly, we added pressure transducers (Baratron 722B,

MKS Instruments. Inc., Andover, MA, USA) to the vWCPC 3789 inlet and the exhaust lines. Finally, when we operated with 1.5 lpm inlet aerosol flow, we blocked the make-up flow port. However, we focused on 0.6 lpm inlet aerosol flow in this study. Therefore, when we operated with 0.6 lpm aerosol inlet flow, 0.9 lpm flow from the exhaust line was filtered, passed through a flow buffer, and then made up the 1.5 lpm vacuum flow. Note that under both operating conditions, the aerosol flow passing the condensation tubing and optical particle detector is 0.3 lpm.

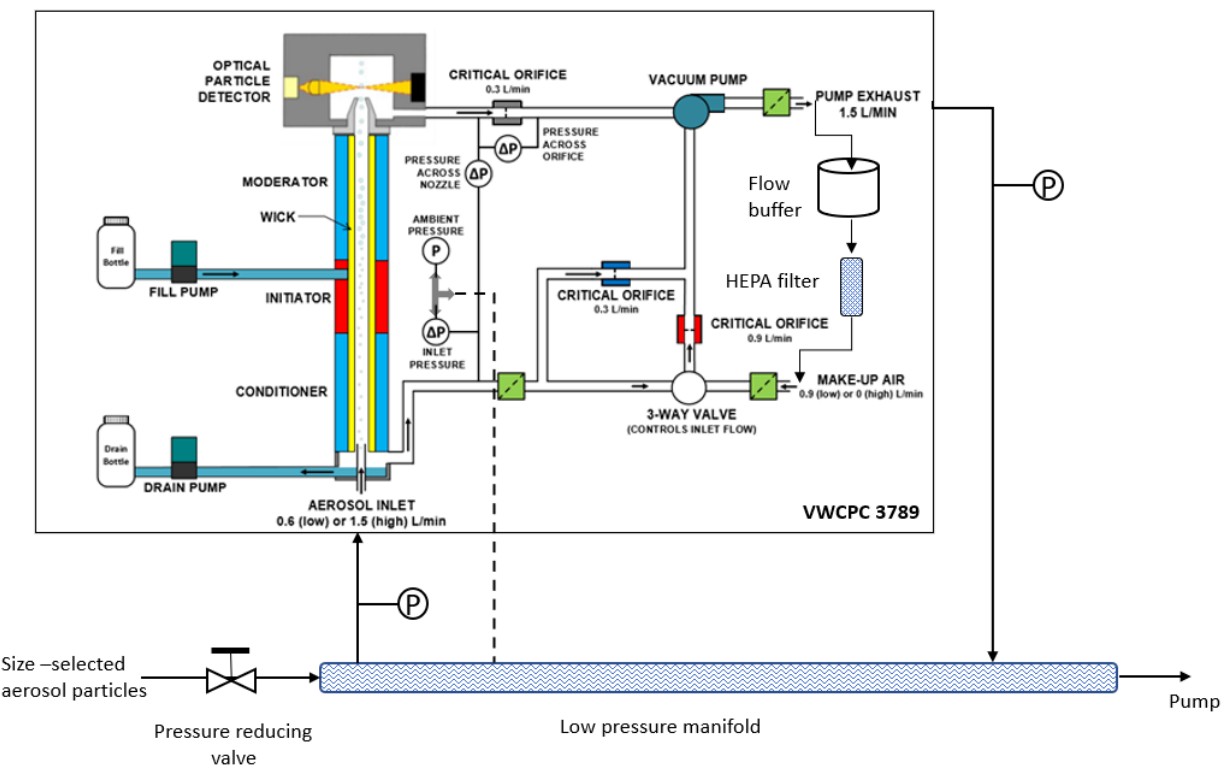


**Figure 1: Schematic of the modified vWCPC 3789 sensors and flow system(TSI Incoporated, 2019).**

## 2.2 Experimental characterization setup

The low-pressure calibration setup of the vWCPC 3789 is shown in Fig. 2. Ammonium sulfate was the primary material for this study and was dissolved into deionized water for aerosol generation using atomization techniques. The water-insoluble chemicals,

such as humic acid and oleic acid, were atomized from a water suspension after at least 10 minutes of ultrasonic aided mixing. The properties of the other tested aerosol particles were included in Table S1. The tested particles selected in this study were commonly

used for the previous CPC characterization (Hering et al., 2014; Hering et al., 2005; Kangasluoma et al., 2017). To increase the aerosol number concentration for particles less than 30 nm, polydisperse ammonium sulfate (AS) aerosols were also passed through a tube furnace generator (Lindberg/Blue, Thermal Scientific, TX, USA) to shift the size distribution a smaller size. Polystyrene

latex (PSL) particles were generated through an atomizer. After passing a dilution system, the aerosol particles were size-selected by a differential mobility analyzer (DMA, TSI 3081) using a soft x-ray neutralizer (TSI, advanced aerosol neutralizer 3088). The operating pressure was reduced by a constant pressure inlet (DMT) to simulate different flight level pressures (500 – 1000 hPa). For the counting efficiency determination, two reference sensors were used. One reference sensor was a CPC 3775 (TSI, butanol-based, 50% cut-off diameter is 4 nm), which was connected directly to the monodisperse flow after the DMA and operated at a 0.3

lpm flow rate under the atmospheric pressure. Its inlet aerosol flow rate was 0.3 lpm. The other sensor, an aerosol electrometer (A.E. 3068B, TSI, Shoreview, MN, USA), was operated parallel with the vWCPC 3789 under low-pressure conditions. Both vWCPC 3789 and A.E. were run at 0.6 lpm inlet flow with matched tubing lengths to ensure equal diffusive particle loss in the aerosol pathway. The DMA's sheath flow was typically 20 lpm, resulting in a sheath to aerosol flow ratio and a non-diffusive mobility resolution of 13.


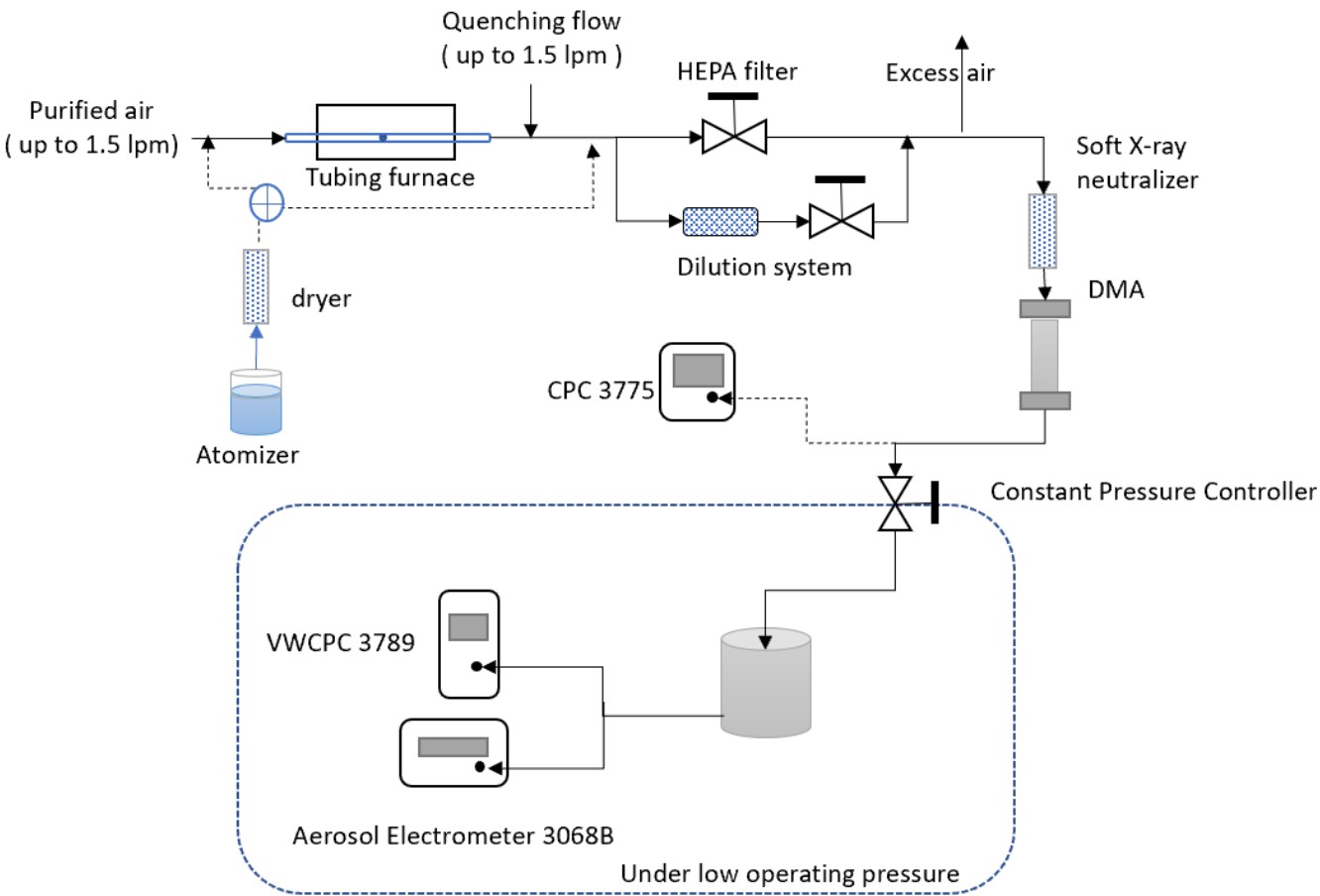

**Figure 2. Schematic of the vWCPC 3789 sensors and flow system under the low-pressure testing.**

**2.3 Numerical simulation**

Numerical simulations of the temperature and humidity profiles and the particle growth were calculated for the geometry of the

vWCPC 3789 described above, using the methods published by Hering's group (Hering et al., 2014; Lewis and Hering, 2013). The assumption and equations used for the simulation are included in the supplemental document. More details are discussed in section

3.2. To summarize the methodology, first, the temperature and humidity profiles were computed using the finite element modeling software - COMSOL Multiphysics® (www.comsol.com. COMSOL AB, Stockholm, Sweden). Next, the particle growth, temperature and humidity profiles were calculated using a numerical model developed by Lewis and Hering (Lewis and Hering, 2013), written in Igor Pro (Wavementrics, Beaverton, OR).

The configuration in Hering's research (Hering et al., 2014) is different from the vWCPC 3789. Hering's WCPC consists of a 4.6 mm I.D. tube extending through a 154 mm conditioner, a 76 mm initiator, and a 100 mm moderator, with a design flow rate of 1.5 L/min. Their simulation results suggest that the three-stage configuration is superior in decreasing the amount of water vapor and lowering the particle loss and variation in detection and collection, avoiding the side-effect of heating the flow. With this "moderated" approach, a short, warm, wet-walled initiator provides sufficient water vapor for activation, followed by a cool-walled moderator for particle growth. A recent simulation study (Bian et al., 2020) confirmed Hering's finding with different temperature settings. In general, enhancing the temperature difference between the initiator and the conditioner can obtain higher supersaturation and smaller activation size.

Furthermore, shifting the 70 °C temperature difference window by decreasing the conditioner temperature (from 9 °C to 1 °C) further reduces the activation size. In addition, Hering's paper pointed out that the final droplet size decreases from around 4.5 μm to 2.5 μm with the increase of the aerosol inlet flow rate from 0.4 lpm to 1.5 lpm. This behavior is consistent with Bian's result, which shows that when the flow rate increased by a factor of 2.5, the final size decreased by 43%.

## 3 Results and discussion

### 3.1 Pressure dependence of the vWCPC 3789 counting efficiency

The CPC counting efficiency is defined by the ratio of the particle number concentration measured by the vWCPC 3789 and the reference particle number concentration measured by the A.E. Multiple charged particles induce a reading error in the reference particle number concentration. This error is negligible for particle sizes less than 70 nm compared to other experimental errors. (Hermann, 2000; Hermann, 2001). For aerosol particles larger than 70 nm, an empirical correction was estimated using the size distribution of the generated aerosol and the aerosol charging distribution (Tigges, 2015, "Bipolar charge distribution of a soft x-ray diffusion charger") and the particle loss through the constant pressure inlet, which can be estimated based on the total number concentration difference measured by the CPC 3775 and vWCPC 3789.

Using the low-pressure testing setup shown in Fig. 2, the counting efficiency of a vWCPC 3789 was measured between 500 hPa to 920 hPa for AS particles of 15 nm, 25 nm, and 100 nm (mobility diameter). The aerosol concentrations in this test were maintained in the range $2\sim4 \times 10^4$ cm$^{-3}$. The obtained counting efficiencies for the manufacturer's 7 nm cut-off setting are shown in Fig. 3. The conditioner, initiator, and moderator temperatures were 30 °C, 59 °C, and 10 °C under this setting.

During the testing, the temperature variations in the conditioner and moderator were less than ±0.5 °C. For the "7 nm" temperature setting, the initiator temperature has a variation of ±1 °C. The y -axis error bar indicates the standard deviation of the counting efficiency averaged over ~5 minutes of sampling time at a 1 Hz sampling rate. Fig. 3 shows that the counting efficiencies decreased with the decrease of the operating pressure around 700 hPa. The decreases became significant and larger than 10% when the operating pressure was lower than 600 hPa.

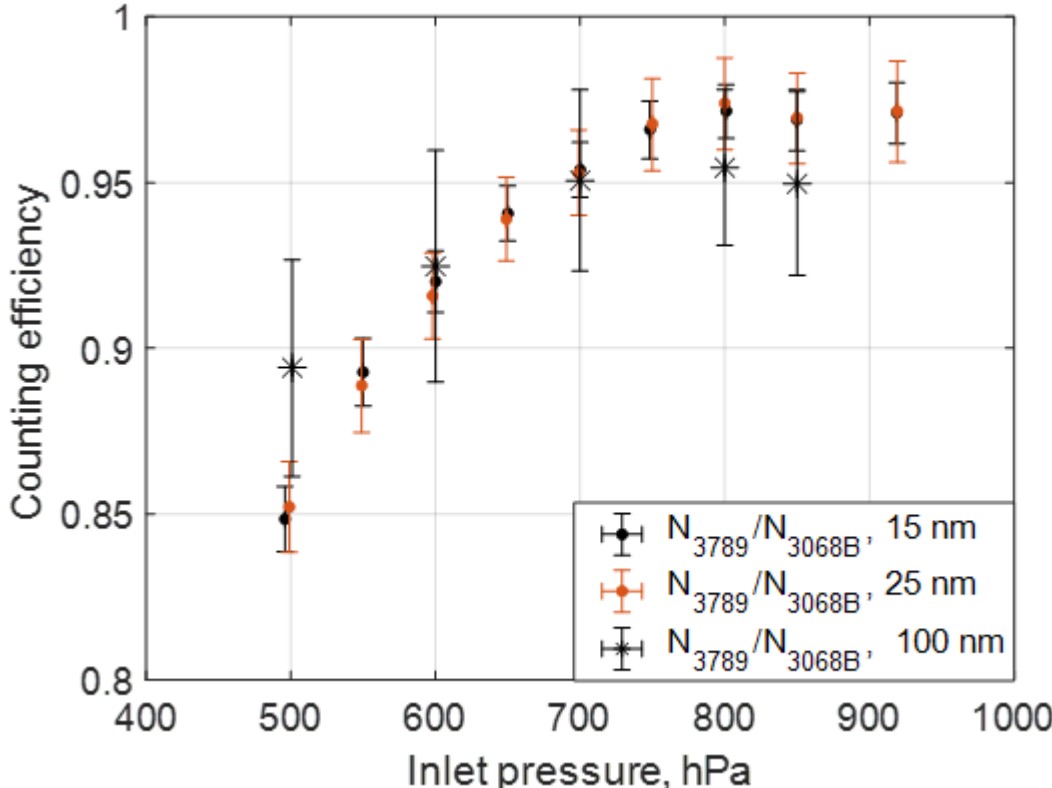

**Fig. 3, vWCPC 3789 counting efficiency as a function of the inlet operation pressure at TSI standard conditions: Tcond = 30 °C ,Tini = 59 °C, and Tmod=10°C. $N_{3789}/N_{3068B}$ is the total number concentration ratio between the vWCPC and the electrometer.**


Based on the $\kappa$-Köhler theory and the supersaturation (relative humidity-1) estimated at the centerline of a three-stage chamber in Fig. S2, ammonium sulfate particles larger than 9.7 nm should all be activated when the saturation ratio is larger than 1.05, even if we assume that the supersaturation profile near the wall has an 85% drop from the centerline condition. Thus, the low counting efficiency under the low-pressure operation condition is not limited by aerosol activation.

Previous simulation studies show that the centerline saturation ratio is not sensitive to the wall temperature of the moderator (Bian et al., 2020; Hering et al., 2014). However, under the low-pressure condition (e.g., 500 hPa), the saturation profile peaked earlier but lower than the saturation profile under the standard condition (1000 hPa), as shown in Fig. S2. Also, decreasing the conditioner temperature (while maintaining the same temperature difference between the initiator and the conditioner) provided higher saturation ratios in the initiator and more water vapor for particle growth, which is also consistent with the previous growth tubing

simulation (Bian et al., 2020). Thus, we examine the temperature effect on the vWCPC 3789 performance under the low-pressure condition in the following section. Furthermore, with the aid of the simulation, we study how to overcome the couniting efficiency decrease experimentally.

**3.2 Simulation-aided investigation of the pressure dependence of the vWCPC 3789 counting efficiency at different conditioner temperatures.**

The simulated saturation profiles of a three-stage growth tube at different operation pressure (1000 and 500 hPa) for three different conditioner temperatures (30 °C, 27 °C, and 24 °C), all with initiator temperature of 59°C and moderator temperature of 10°C, are presented in Fig.4. The predicted droplet size evolution along the growth tube of this vWCPC 3789 at different operation pressures (1000 and 500 hPa) under three conditioner temperatures (30 °C, 27 °C, and 24 °C) is included in Fig. 5. In most water-based CPCs,

the aerosol particle growth is initiated by the temperature rise from the conditioner to the initiator. The growth rate is modulated by the temperature decrease from the initiator and the moderator. The maximum saturation ratio calculated at the centerline usually occurs downstream of the initiator exit due to the water vapor diffusion delay. However, in Fig. 4, we observed a double-peaked saturation ratio profile appeared for three different conditioner temperature settings under both operation pressures (1000 and 500 hPa). The maximum saturation peak occurs inside the moderator under most simulated conditions, except when the operation pressure decreases to 500 hPa at the conditioner temperature of 24 ℃, the highest saturation peak occurs inside the initiator. Under the standard pressure, this configuration has the advantage that the peak extends close to the wall, resulting in high counting efficiency and a sharp lower cut-off size, as shown in Fig. 5. With the decrease of the operating pressure, the saturation peaks showed a substantial reduction, which was also associated with both the lower cut-off size increasing in Fig. 4. and the growing droplet size decreasing in Fig 5. When the conditioner temperature is 27 ℃ or 30 ℃, with decreasing of the operating pressure from 1000 hPa to 500 hPa, one 8 nm seed particle grew to a smaller size (~40% reduction in the droplet diameter), no matter the particle entered the growth tube in the centerline or near the wall. The seed particles entering the centerline of a growth tube got activated in the initiator. Delay of the activation occurred for the seed particles entering the growth tube away from the centerline, and those particles at 75% of radius started growing in the moderator.

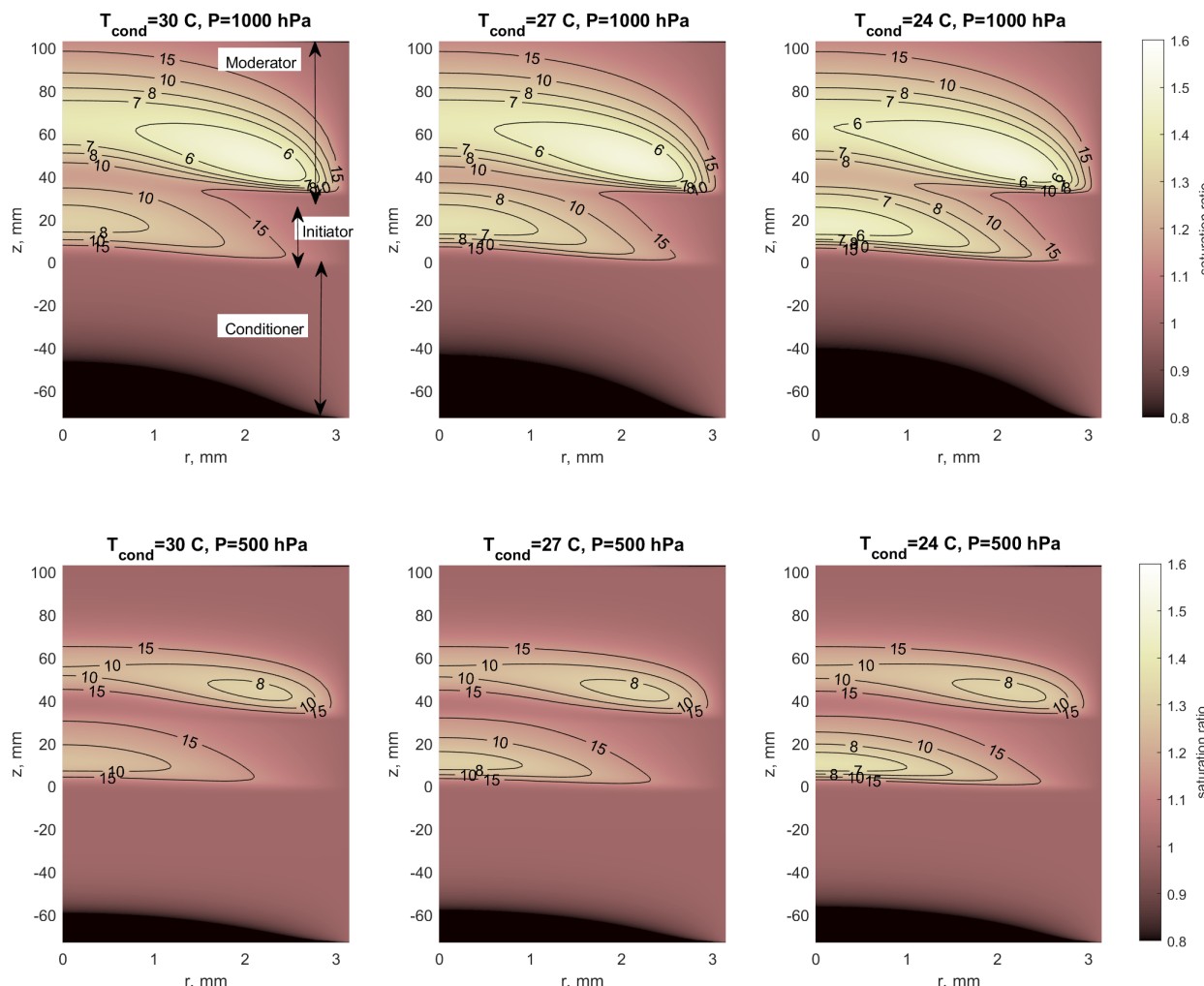

**Fig 4. Simulation of the CPC3789 saturation ratio profiles at 1000 hPa (upper panel) and 500 hPa (bottom panel) under the different conditioner temperatures (30 ℃, 27 ℃, and 24 ℃) with the initiator temperature at 59 ℃, and the moderator temperature is 10 ℃. The color bar indicates the humidity (the saturation ratio) change inside the three-stage growth tube. The contour line indicates the saturation ratio necessary to activate 6, 7, 8, 10, 15 nm seed particles. r is the distance from the centerline. z is the distance from the entrance of the initiator, which means that the conditioner is from -70 to 0 mm.**

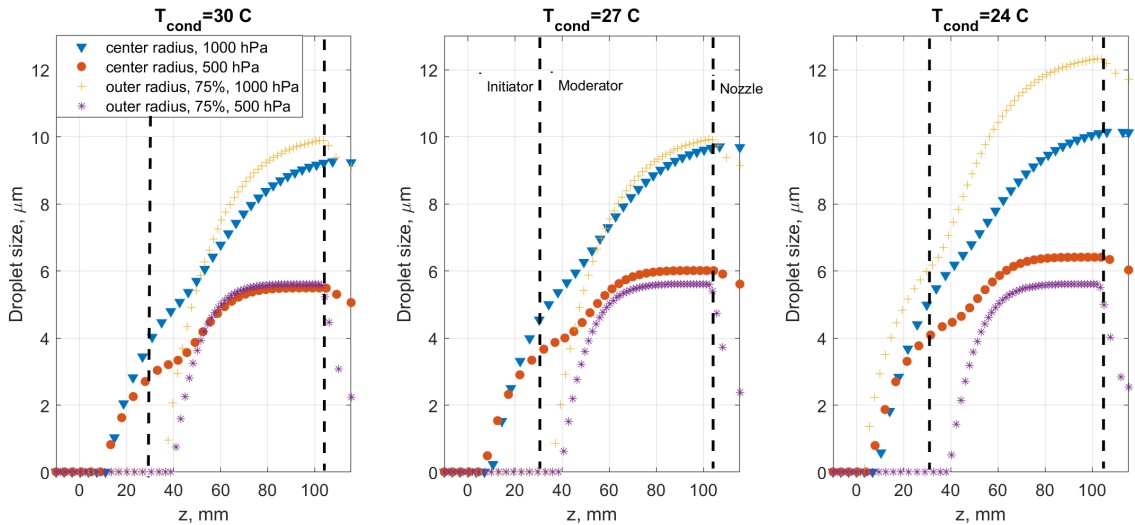

**Fig. 5. Predicted droplet size evolution along the growth tube (at centerline or 75% of the inner tubing radius) of the vWCPC 3789 under the different conditioner temperatures (30 ℃, 27 ℃, and 24 ℃) with the initiator temperature at 59 ℃, and the moderator temperature is 10 ℃. Starting particle size 8 nm. The dashed lines indicate the starting and the ending locations of the moderator.**

Guided by the above analysis and the observations, we maintained the temperature settings in the initiator at 59°C and moderator at 10°C and varied the conditioner temperatures by 3 ℃. The aerosol particle concentrations for 100 nm size-selected particles were maintained at around $6 \times 10^3 (\text{cm}^{-3})$ in this study to avoid the concentration effects (more discussion in section 3.3). Under each conditioner temperature, we examined the impact of the operating pressure on the aerosol particle growth inside the three-stage growth tube through the simulation (shown in Fig. 4, 5, and S2) and on the vWCPC 3789 counting efficiency change through the experiment (as shown in Fig. 6). When the conditioner temperature was set at 30 ℃ or 33 ℃, the counting efficiency drops as the inlet pressure is lowered from 1000 hPa to 500 hPa. However, when the conditioner temperature was decreased to 27 ℃ or 24 ℃, we did not observe a noticeable decrease in the counting efficiency with the operating pressure decrease.

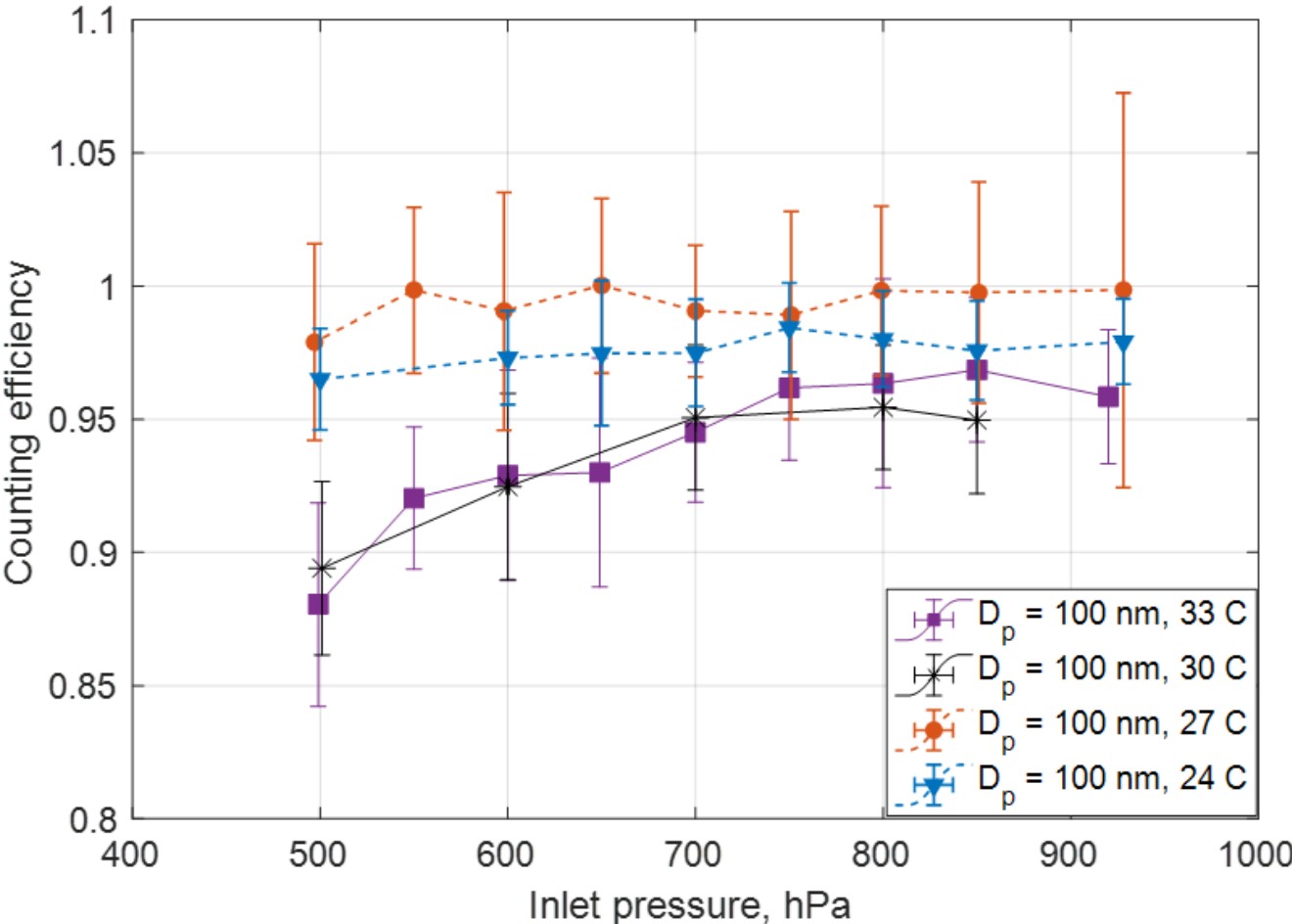

Fig. 6, vWCPC 3789 counting efficiency as a function of the inlet operation pressure at four conditioner temperatures of $T_{cond}$ = 24, 27, 30, 33 °C (Tini=59°C, Tmod=10°C).

The experimental results suggest that increasing the temperature difference between the conditioner and the initiator affected the vWCPC 3789 counting efficiency. As shown in Fig. 6, the counting efficiencies of the $T_{cond}$=27 °C setting were maintained close to 1 when the operating pressure dropped to 500 hPa. Although we presented data with 100 nm particles in Fig. 6, we observed a similar trend with particles down to 15 nm (Fig. S3). This observation suggests that the low counting efficiency observed in Fig. 3 was not mainly caused by the particle loss inside of the instruments. Fig. 5 suggests that although the 2nd saturation peak in the moderator activated the seed particles and is capable of maintaining droplet growth, about 10% of particles may not grow large enough to be detected in the optical chamber. Therefore, the counting efficiency is more susceptible to the 1st peak of the supersaturation profile in the initiator. In addition, the absolute saturation also set the threshold for successfully operating this vWCPC 3789 under lower pressure. Combining the observations from Fig 6 and Fig S3 and the simulation in Fig. 4 and 5, when the simulated saturation is over 1.3, the counting efficiency maintained close to 1 for aerosol particles larger than 8 nm under low-pressure conditions (500 – 920 hPa). We also noticed that the counting efficiency curve is slightly lower when operating the conditioner at 24 °C than when the conditioner was set to 27 °C. That is possibly due to the high supersaturation profile inside the three-stage tube, leading to larger droplets, especially close to the wall and increased loss at the tubing wall and through the focusing nozzle.

**3.3 Particle concentration and pressure effects on droplet size and vWCPC 3789 counting efficiency**

In typical operation, the manufacturer reports that the single-particle counting concentration is maintained up to a concentration of $2 \times 10^5$ (cm$^{-3}$). The reported "pulse height" by a vWCPC 3789 is not the pulse height value produced by the light scattering signal. It is a calculated parameter, which indicates the fraction of the particle population generating an acceptably high pulse. The manufacturer's manual states that the pulse height is above 0.9 for moderate concentrations (~10-5,000 cm$^{-3}$). Thus, we examined if this concentration threshold would hold under low-pressure conditions. For 100 nm particles, we observed that the pulse height decreases with the decrease of the operating pressure, as shown in Fig. 7(a). When the operation pressure decreased from 550 hPa to 500 hPa, the pulse height decreased from above 90% to around 80%, as shown in Fig 7 (a). Meanwhile, for 100 nm aerosol particles, the threshold concentration ($N_{mea}$, under 500 hPa) for a 10% reduction with the pulse height values was about $1 \times 10^4$(cm$^{-3}$), as shown in Fig. 7 (b). When the aerosol concentration was larger than $2 \times 10^4$ (cm$^{-3}$), decreasing the pressure affected the measured aerosol concentration more significantly, as shown in Fig. S6(b). Comparing the pulse height curves, while the conditioner temperature operated at 24 °C and 30 °C, Fig. S6(a) shows that the 30 °C case suffered more water vapor shortage while decreasing the operating pressure. Additionally, Fig. S6 shows there is no significant difference between the measured 10% reduction threshold between 20 nm and 100 nm particles when the particle concentration is less than $1 \times 10^4$ (cm$^{-3}$). This observation was consistent with the simulated 10% reduction of s and $D_p$ happened when the $N_{7\mu m} \sim 8.5 \times 10^3$(cm$^{-3}$) as discussed in the supplemental material (shown in Fig. S4). Moreover, it indicates that we can monitor the pulse height value to detect the undercounting issue. When the pulse height was less than 80%, the measured aerosol concentration by vWCPC 3789 was about 10% less than the aerosol concentration measured by the electrometer.

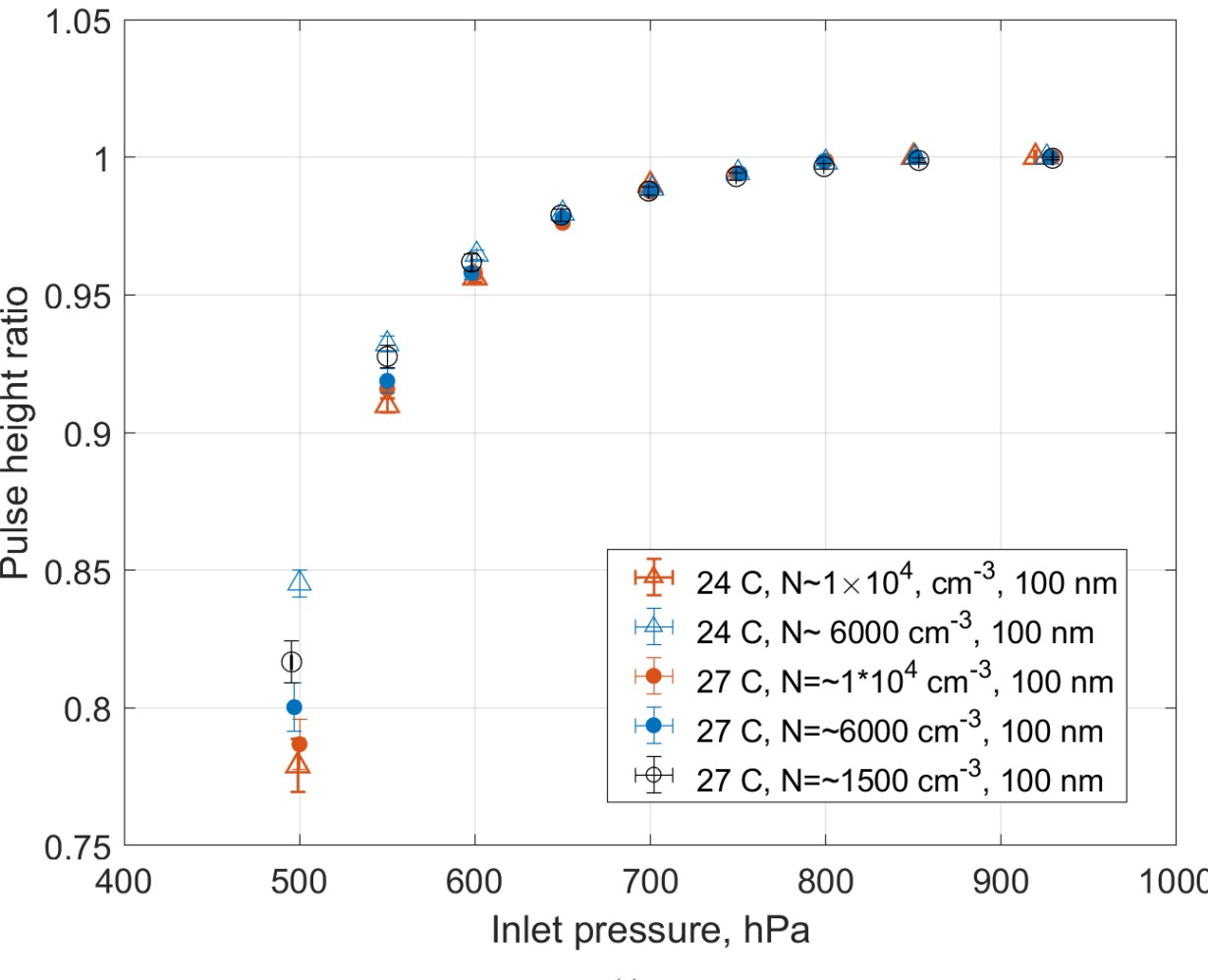

285

(a)

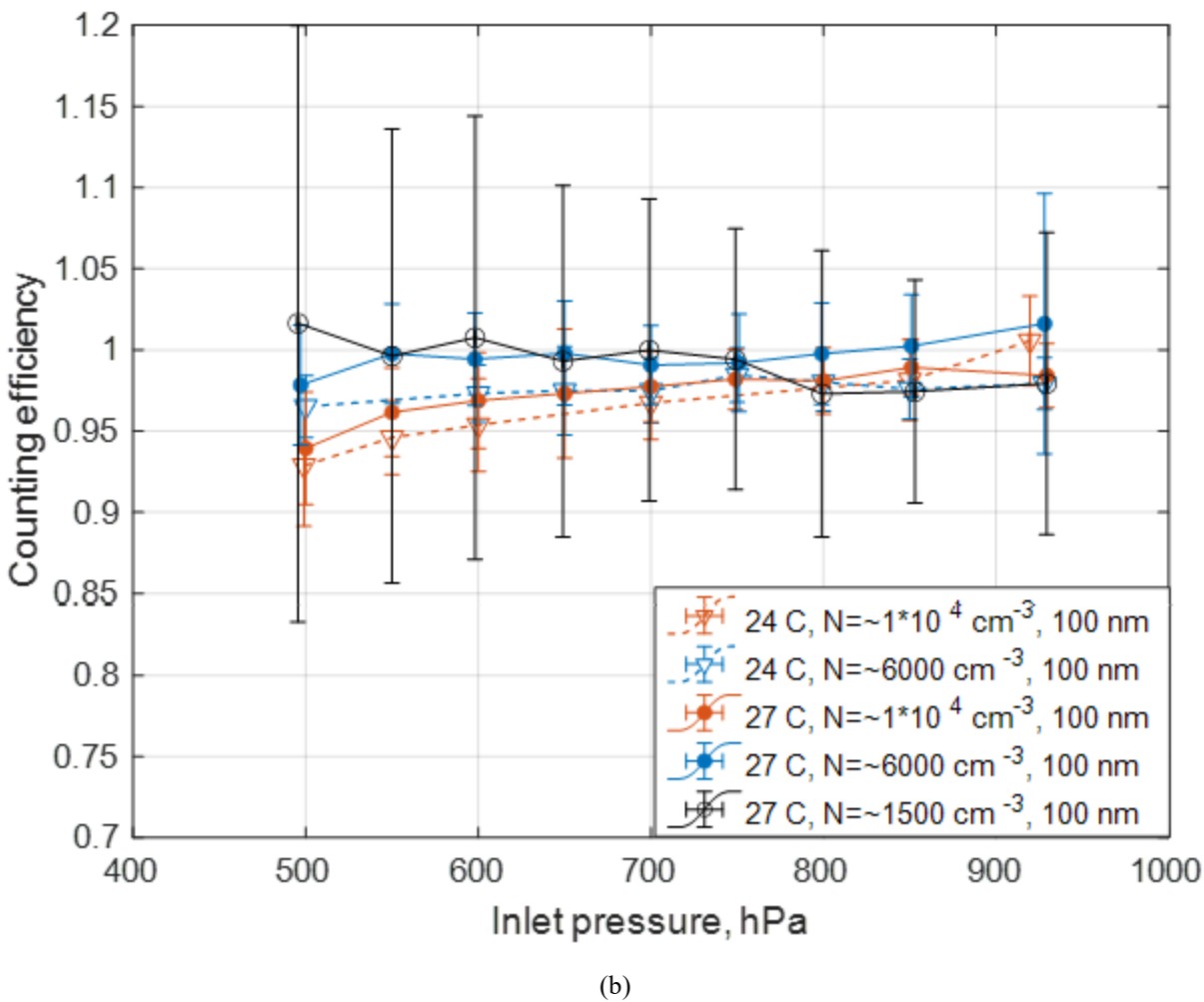

(b)

Fig. 7. The water depletion due to the aerosol number concentration, illustrated by (a) the pulse height generated in the optical detector, (b) the counting efficiency as a function of the inlet pressure. Results are shown with the conditioner temperatures set at 24 °C and 27 °C.

**3.4 Effect of particle chemical composition on the vWCPC 3789 counting efficiency**

We examined the counting efficiency of the vWCPC 3789 using aerosol particles with different chemical compositions and water solubility, as shown in Table S1. During this test, 100 nm aerosol particles were atomized and dried from the ammonium sulfate, sucrose, humic acid and PSL solutions or suspensions before entering the DMA, as shown in Fig. 2. We chose two types of 100 nm aerosol particles: water-insoluble particles, such as oleic acid, humic acid particles and PSL, and highly hydrophilic ammonium sulfate and sucrose particles. Running the 100 nm particles under different operating pressures, we got the counting efficiencies close to 1 for PSL, humic acid and AS particles, as shown in Fig. 8. However, the counting efficiencies of the oleic acid and PSL particles showed more significant uncertainty than that of the ammonium sulfate particles. We could not achieve a reliable curve of the oleic acid particles because oleic acid particles evaporated under low-pressure conditions and caused a significant variation in the number concentration and size distribution of size-selected particles. We noticed that the PSL particle curve has more substantial variation compared to the other aerosol particles. The particle surface is very hydrophobic, and the droplet growth process will be affected by the remaining water or surfactant on the particle surface. The counting efficiency of humic acid particles was slightly lower than that of AS aerosol particles, which could likely be explained by the light-absorbing properties of humic

acid. The counting efficiency of sucrose is significantly lower than the counting efficiency of AS (10% lower). This observation is consistent with the previous study (Hering et al., 2017). One possible explanation is that the chemical similarity between the seed particle material and the working fluid also affects the detection efficiency of vWCPC 3789 (Wlasits et al., 2020).

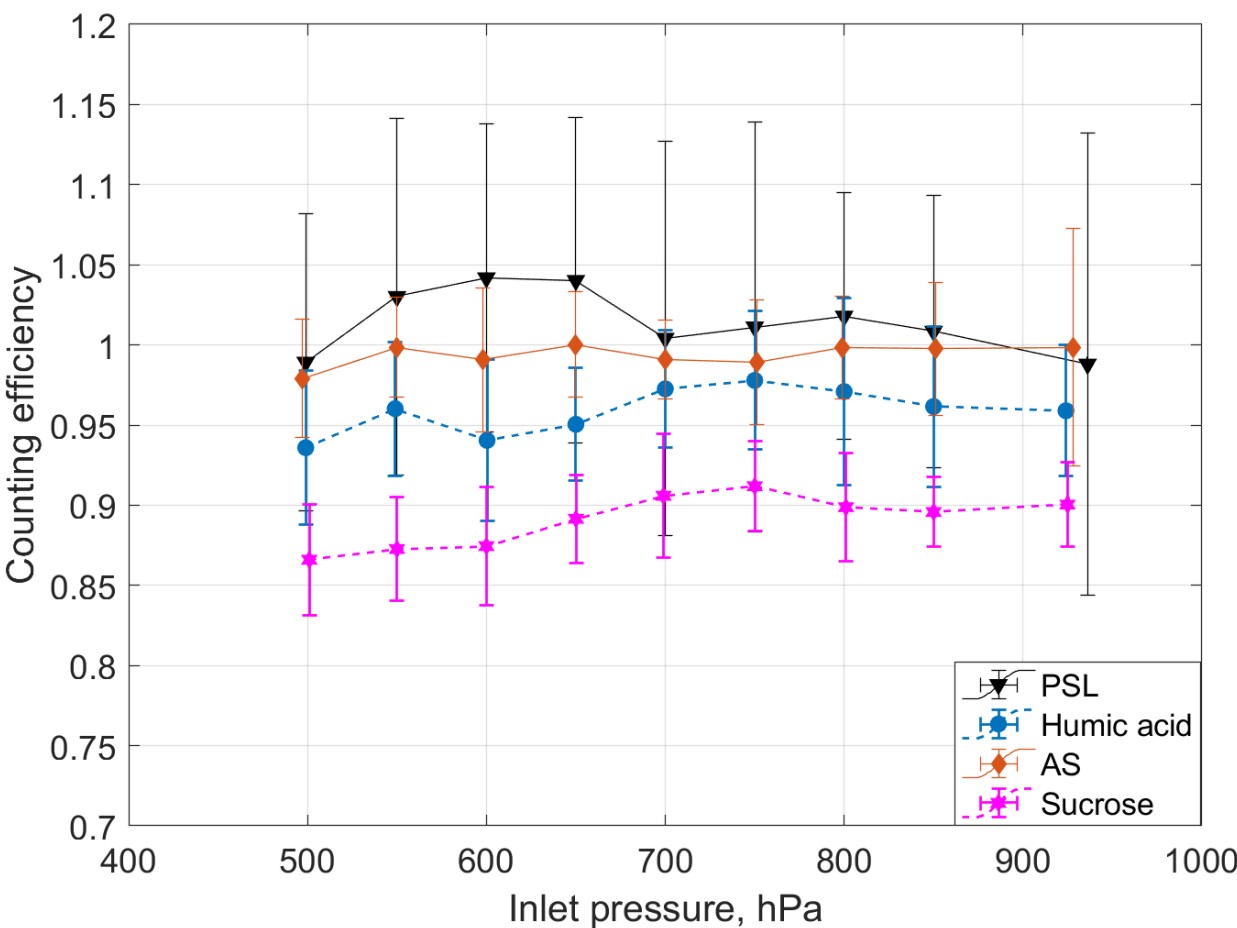

**Fig. 8, vWCPC 3789 counting efficiency at different operating pressure with four different types of 100 nm aerosol particles (PSL, humic acid, AS, and sucrose), when the temperature conditions are Tcond = 27 °C and Tini = 59 °C.**

### 3.5 Temperature dependence of the vWCPC 3789 cut-off size

For the butanol-based CPCs, such as TSI CPC 7610, the CPC cut-off size is strongly influenced by the temperature difference

between the saturator and condenser (Hermann and Wiedensohler, 2001; Kangasluoma and Attoui, 2019). Typically, the cut-off size decreases with the increase of the temperature difference (Hermann et al., 2005; Hermann and Wiedensohler, 2001). We observed that this trend held well for the vWCPC 3789 using AS particles, as shown in Fig. 9. The temperature effect on the counting efficiency of the vWCPC 3789 under low-pressure conditions is also presented in Fig. 9. With the decrease of the conditioner temperature, the temperature difference between the initiator and the conditioner increases. As expected, the cut-off

size slightly moved to lower than 7 nm. However, with the decrease of the operating pressure, the cut-off size increased slightly. Under two conditioner temperatures (24 °C and 27 °C), the cut-off size was maintained between 6-8 nm, even with the pressure dropped to 500 hPa. Thus, both settings are suitable for airborne operations up to 5.5 km with a 7 nm cut-off size. Note that the counting efficiency curve from TSI at 30 °C was derived and fitted using AS particle classified by a custom-made different mobility analyzer (using an aerosol to sheath flowrate ratio of 1:100) under a near-sea-level pressure (Wlasits et al., 2020).

As a  result of operation over a much wider pressure range, the cut-off curve derived by this study is less sharp than for the TSI
standard settings.

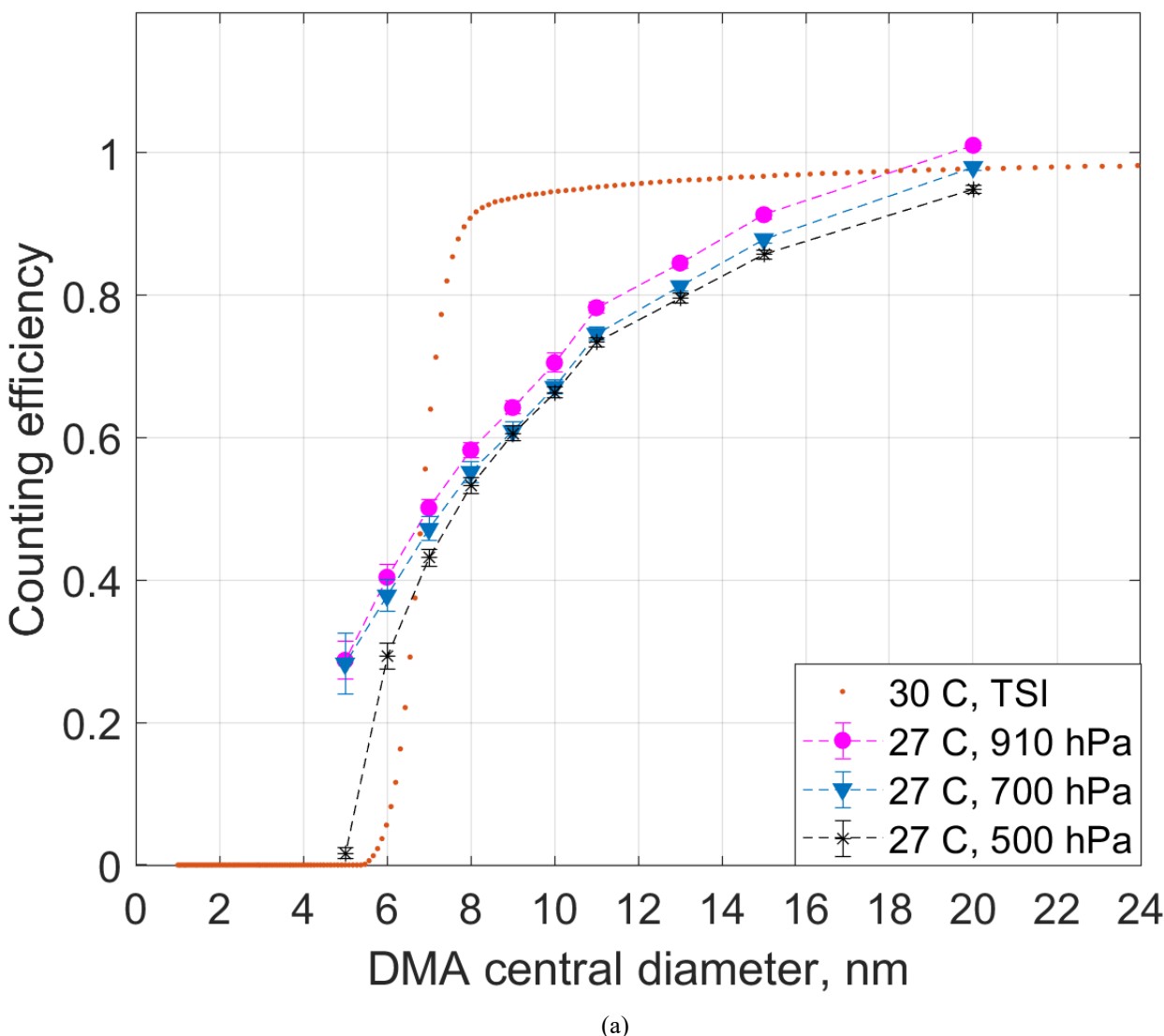

(a)

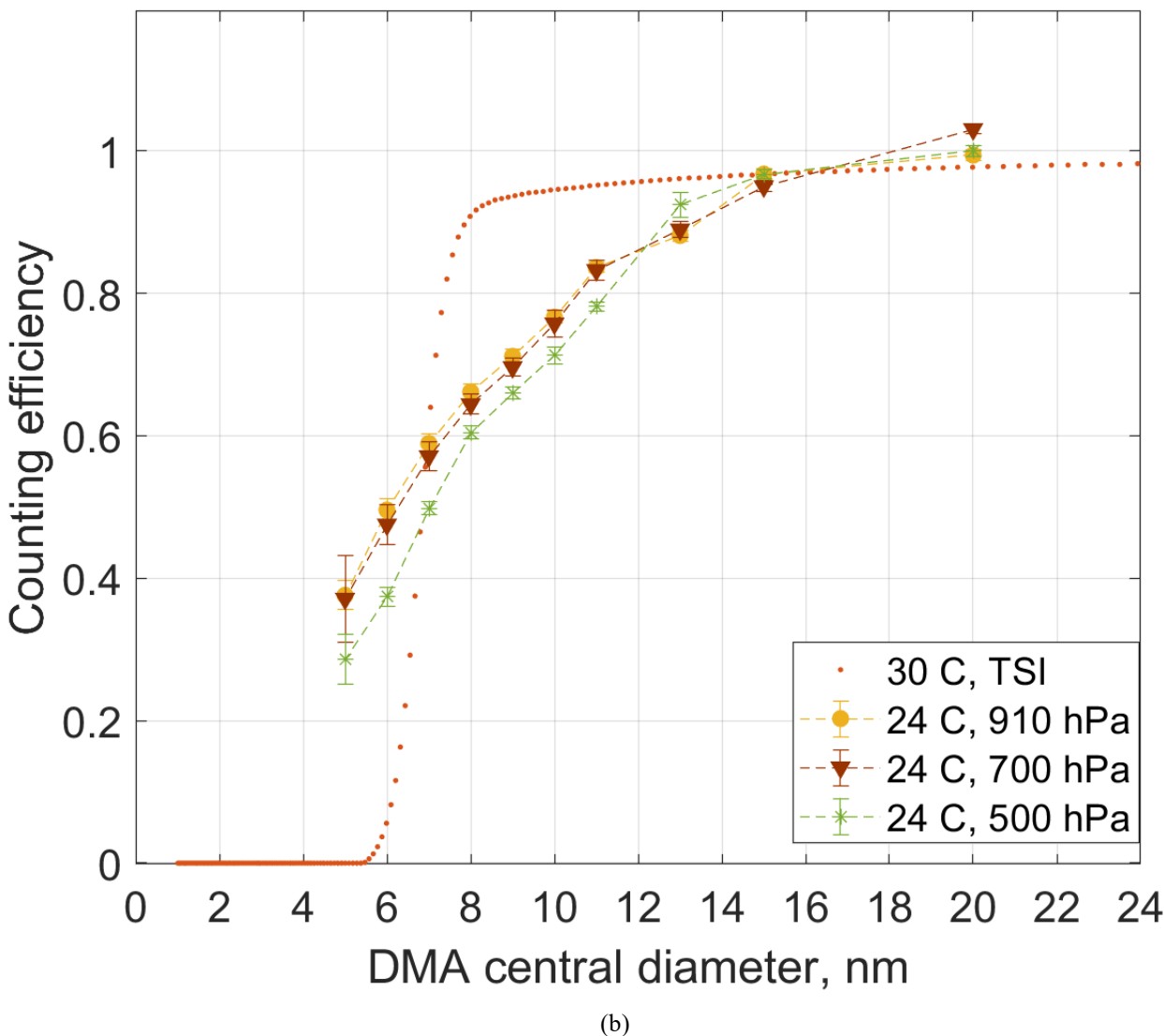

(b)

**Fig. 9, vWCPC 3789 counting efficiency changes as a function of the different operating pressures (500, 700, and 910 hPa) using AS particles, when the initiator temperature is 59 °C, and the moderator temperature is 10 °C, at two different conditioner temperatures (a) Tcond = 24, (b) Tcond = 27 ℃.**

## 4 Conclusions

This study discusses the modification and characterization of the versatile water CPC (TSI 3789) operating under low-pressure conditions. A commercially available vWCPC 3789 was modified to report the inlet operating pressure during the airborne operation. The vWCPC 3789 counting efficiency was characterized as a function of operating pressure (500 – 920 hPa) for different conditioner temperatures (24 – 33 °C) with the factory settings for the initiator and moderator temperatures (59° and 10°). The

340 vWCPC 3789 with all manufacturer settings (i.e., conditioner temperature of 30 °C) worked as expected under the standard ambient condition (i.e., 1 atm). However, under low-pressure conditions, the counting efficiency of the vWCPC 3789 operated with the factory settings decreased with the decrease of the operating pressure, especially when the operating pressure was below 700 hPa. Although not being able to fully explain the decrease of counting efficiency, numerical simulation and the dimensionless analysis show that the peak saturation ratio at 500 hPa is significantly lower than that at 1000 hPa but occurs closer to the entrance of the

initiator. At the same operation pressure, the simulated peak saturation ratio increases with decreasing conditioner temperature. Aided by the simulation results, we examined the effect of conditioner temperature on the counting efficiency and identified an optimal conditioner temperature setting of 27°C for operating the vWCPC 3789 over a range of pressure levels. Additionally, decreasing the conditioner temperature to 27°C did not significantly change the lower cut-off size – 7 nm for the vWCPC 3789. Thus, for the airborne operation down to 500 hPa (~ 6000m above sea level), we recommend operating the vWCPC 3789 with the conditioner temperature setting of 27°C. We also observed that the conditioner temperature has a more pronounced effect on the vWCPC's counting efficiency curve than the operating pressure. A simplified water depletion estimation suggests a 10% reduction of the saturation ratio (s) and the droplet diameter ($D_p$) under a pressure of 500 hPa when $N_{7\mu m} \sim 8.5 \times 10^3 (cm^{-3})$. The impact of aerosol number concentration on the pulse height reported by the vWCPC 3789 was examined. Similar to the counting efficiency, the pulse height exhibits a decreasing trend with decreasing operating pressure. When the pulse height was larger than 80%, and the particle concentration was less than $1 \times 10^4$ ($cm^{-3}$), a 10% reduction in the measured concentration was observed for 20 nm and 100 nm particles. This observation suggests that the reported pulse height could be used to monitor the potential bias caused by high particle concentration. The chemical composition of aerosol also contributes up to 20% uncertainty in the counting efficiency of CPC, and this uncertainty shows no significant trend with the operating pressure changes when the CPC is operated with the conditioner temperature at 27°C.

The main advantage of vWCPC 3789 is the non-toxic and non-flammable working fluid. Additionally, water has a smaller calculated Kelvin diameter than other working fluids, which means a water-based CPC could lead to a lower size detection limit (Magnusson et al., 2003a). However, for the airborne operation, the elevation will reduce the CPC operating pressure, which limits the highest temperature we could choose for the initiator to push the detection limit lower. Thus, further studies are needed to study how to reduce the lower detection limit of vWCPC 3789 under various ambient pressures.

**Acknowledgments:** This work has been supported by the Office of Biological and Environmental Research (OBER) of the U.S. Department of Energy (DOE) as part of the Atmospheric Radiation Measurement (ARM) and Atmospheric System Research (ASR) Programs. Battelle operates the Pacific Northwest National Laboratory (PNNL) for the DOE under contract DE-A06-76RLO 1830. We sincerely appreciate the valuable discussion with Andrea Tiwari (TSI), Oliver Bischof (TSI) and Justin Koczak (TSI).

**Data availability**
The CPC data in the study are available upon reasonable request to Fan Mei (fan.mei@pnnl.gov).

**Author contributions**

F.M., M.S.P., S.H., and J.W. designed the research. F.M. carried out the measurements. F.M. led the analyses, and S.S. and G.L. led the simulation. F.M. led the writing, with significant input from S.H. and J.W. and further input from all other authors. B.S. and J.T. acquired the financial support for the project leading to this publication. M. H. provided suggestions during the experimental design. S. H., J.W. and B. S. provided suggestions on the revision.

**Competing interests**

The authors declare that they have no conflict of interest. Susanne Hering and Maynard Havlicek have a commercial interest in the success of the vWCPC instrument.

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
