# Peer review of "Simulation-aided characterization of a versatile water condensation particle counter for atmospheric airborne research"

_Atmospheric Measurement Techniques, 2021_

## Author Comment (AC1)

OVERALL COMMENT: The manuscript describes the results of the study on the performance of a commercially available versatile water-based condensation particle counter, vWCPC TSI 3789, at a range of pressure conditions down to 500 hPa. The reason behind it is to gain a better understanding of the instrument behavior in potential future airborne applications. The authors investigate the vWCPC counting efficiency and the cut off size, as well as the effect of the conditioner temperature, particle material and particle number concentration at a range of pressure conditions between 500 and 1000 hPa. This is done using both laboratory experiments and numerical simulations.

The topic fits well under into the scope of the AMT and I recommend the manuscript to be accepted after the following minor points have been addressed.

**Response: We sincerely appreciate the comments and suggestions from our reviewer. Thank you very much for considering the publication of our manuscript. We address your specific comments below (also in blue). The line number is corresponding to the change-tracked version.**

**SPECIFIC COMMENTS:**

Line 10-20: Abstract: The authors may add more information such as the d50 for the vWCPC used in this study; a range of pressure settings investigated; a range of particle number concentrations, and particle material used.

**Response: Thank you very much for this constructive suggestion. We revised the abstract accordingly.**

Line 14: authors may want to add e.g.: "under a large range of ambient pressure conditions"

**Response: we revised the sentence in line 15-16.**

Line 19: "chemical composition" of particles?

**Response: we revised the sentence in line 21.**

Section 1: Introduction: The authors mention that non-water-based CPCs have been investigated under low pressure conditions, however, the discussion on the outcome of these studies as well as deployment of these on aircrafts is not mentioned. It would be interesting to discuss why such studies were done more often for butanol-based CPCs or other CPCs rather than for WCPCs? Win general, what are the main challenges for the implementation of CPCs on aircrafts, and what are specific challenges for WCPCs? Are health and safety concern the only advantage of the vWCPC over other CPCs in such applications? Some more discussion on that would add value to the manuscript. Although the manuscript talks about the history of CPCs and wCPCs, the literature review says nothing about the state-of-the-art CPCs or CPC-based systems currently operating on aircrafts. I think more information that is currently provided could be discussed in the context of low-pressure applications. It would be interesting to see what is currently the state-of-the-art instrumentation (i.e. CPCs) - a good example could be NMASS instrument by Williamson et al. 2018.

**Response: we revised section 1 and added more discussion on the other airborne CPCs (line 61-70) and the challenges for wCPCs (line 84-85).**

Line 23-44: The authors describe the history of and some new developments in CPC instrumentation. Have any of these CPCs been used on aircraft platforms or was their performance studied under low pressure conditions? If yes, that would be interesting to highlight.

**Response: Thank you for this suggestion. We add a paragraph in line 61-70 mentioned two aircraft operating CPCs.**

Line 56: delete the extra bracket

**Response: Deleted the extra bracket. Thank you for catching that.**

Line 71: The authors mention "comparable performance" – in terms of?

**Response: we revised in line 115-116. "in terms of the cut-off size and detection efficiency."**

Line 71-75: The authors mention that non-water-based system have been characterized under low pressure conditions. I am currently missing information on the outcome of these studies.

**Response: we added the additional information of the non-water-based system in lines 61-70 and 84-85.**

Line 78: "pressure dependency of the counting efficiency" - what were the results?

**Response: we revised the sentence in line 122, "... and lower cut-off diameter of the CPCs usually increased with the decrease of the operating pressure"**

Section 2.1 Instrument modification: The authors may consider "instrument description and modification" instead. Additional information on the instrument description could be helpful (such as the cut off size for certain particle material and temperature settings (e.g. default), detection of the maximum number concentration, default settings in terms of temperatures, new design of the growth tube.

**Response: we revised the section 2.1 title and line 141-143.**

Fig1: indicating which modifications were made as compared to the default unit could be helpful here?

**Response: the 0.9 lpm flow from the exhaust line was filtered, passed through a flow buffer, and then made up the 1.5 lpm vacuum flow.**

Line 99: "each vWCPC" - how many were used in this study?

**Response: we tested three vWCPC in this study, and they had similar performances. However, we reported one vWCPC result in this manuscript. Thus, we revised the sentence to "the testing vWCPC".**

The authors may want to mention somewhere in the manuscript whether the vWCPC was operating with or without water fill bottle under low pressure conditions. This is usually what's being done for butanol-based systems (butanol fill bottle stays disconnected/autofill deactivated). If that was the case and vWCPC was operating without fill water bottle connected, please provide additional information e.g for how long, and how was it ensured that such configuration did not compromise the performance of the vWCPC. What was the stability of the detection efficiency when operating in such setting? This could be added to the result section or supplemental material.

**Response: the vWCPC was operated with a water fill bottle connected. We added this information in line 155-156.**

Line 113: Information could be added on: the size ranges of particles used in this study, the particle material, range of pressures and number concentration investigated.

**Response: we used the size-selected particles in this study. The particle material properties are listed in table S1. We revised line 169 -190. The range of operation pressure is in line 193. The number concentrations for each section were updated in the following sections.**

Line: 118: There is no information provided on the CPC 3775 used in this study. I suggest adding information on d50, particle material, and working fluid. How does the response of these two models (vWCPC and 3775) compare without the constant pressure inlet? Maybe a plot could be added to a supplemental file?

**Response: we revised section 2.2 and added more information about 3775 in line 194-195.**

Line 123- 129: seems like a mix of results or methods that doesn't really fit under the section "Experimental characterization set up". Please revise.

**Response: we revised this section 2.2 and moved some contents to the following sections 2.3 (line 204-207).**

Fig.2: are these two dashed arrows that are going out of the atomizer correct? Please double check. Was either the atomizer used or tube furnace? If yes, then one line is incorrect. "Inlet pressure controller" or as in text "constant pressure inlet". Try using same name in both places to be consistent. Did you use a drier? I do not see it in the schematic. If not, please discuss why that was not the case.

Response: we revised Fig2 and section 2.2. Ammonium sulfate was the primary material for this study and was dissolved into deionized water for aerosol generation using atomization techniques. To increase the aerosol number concentration for particles less than 30 nm, polydisperse ammonium sulfate (AS) aerosols were also

**passed through a tube furnace generator (Lindberg/Blue) to shift the size distribution a smaller size.**

Line 132-134: The information on the particle material could be somehow linked with line 113. "this study"? Please mention particles with various composition used and their size ranges. How the set up used in this study differs from other studies e.g. where butanol-based CPCs were used.

**Response: we revised section 2.2 in line 155 - 160.**

Section: 2.3 Numerical simulation: Currently it is unclear whether the approach described here is the one taken in this study, or it was used in previous studies. Please make links between information provided from other studies with the current study. The authors refer to Hering's research – what does that mean? How does the configuration and mentioned dimensions compare to the one of this study?

**Response: we revised section 2.3 to explain our approach first, then summarized the current simulation findings.**

Section 3: Results: the authors may consider: "Results and discussion"

**Response: we revised it to "Results and discussion"**

The authors divide subsections into modelled (3.2) results but under this section one finds experimental results too (e.g. Fig.6). Please revise your subchapter titles/division.

**Response: we tried to show how the modeled result guides the experiments. We revised the section title to avoid confusion.**

Line 159-160: the authors could combine these two sentences into one

**Response: we revised line 160-190.**

Line 157-158: please specify particle material and include "in diameter". Why these sizes were used?

**Response: we revised line 285. We tested more than three sizes (up to 450 nm). However, larger size particles selected by the DMA techniques are subject to more particle loss through the constant pressure controller. Thus, we presented three of sizes here (15, 25 and 100 nm).**

Line 164: "lower than 500 hPa" - data below 500 hPa is currently not in the plot. However, the authors discuss the behavior below 500 hPa. Please revise this statement to match the data presented or add the data that supports that. The counting efficiency start decreasing below ~700 hPa. How does this compares with the results of the previous studies. Any potential explanation? There's no 100 nm data point at 920 hPa. Please double check or state the reason.

Response: sorry that it is a typo. It should be "lower than 600 hPa". For 100 nm particles, we tested fewer pressure conditions to confirm the trend on the same day we tested 15 and 25 nm particles.

Line 198: please rewrite: "one 8 nm see particle grew to a smaller size"

**Response: we revised "one 8 nm seed particle grew to a smaller size"**

Fig. 5. add the meaning of dashed lines in the figure caption.

Response: The dashed lines indicate the starting and ending locations of the moderator.

Line 214: this subsection is called "Modelled...", however it contains both simulated and experimental results. Please revise.

**Response: we revised to "Simulation-aided pressure dependence study of the vWCPC counting efficiency at different operating temperatures"**

Line 217: add information for which particle size. Why did you choose 100 nm and not 15 or 25nm in diameter here?

Response: we revised in line 368 to add 100 nm and added one more sentence in line 367-368. We also tested aerosol particles with other diameters, and they showed similar counting efficiency trend changes as Fig 6. When we observed the counting efficiency decreases, we didn't know how much change the supersaturation profile inside of the instruments might have. Therefore, we choose 100 nm to ensure the aerosol particle activation is not affected by the supersaturation decreases.

Line 234: "low pressure conditions" – please be more precise

**Response: we revised line 392.**

Line 235: the authors mention "saturation ratio over 1.3 and particles larger than 15 nm" however this is not what is presented in these figures (4,5,6). Please review the sentence/figures.

**Response: we revised in line 384- 385 and line 390.**

I am curious to know whether you tried any pressure setting below 500hPa in your study? If yes, what was the lowest pressure and what were the results?

Response: the lowest setting we tried is 300 hPa. Above 500 hPa CPC, the performances are consistent among the three vWCPCs we tested. The two of three worked well to 300 hPa with the moderate setting changed to 27 C, but not the first loaner one, which I couldn't further evaluate after returning the loaner.

Line 241: although it may be obvious to those who worked with CPCs, please briefly explain what the pulse height is

**Response: we revised in line 400 and added "The reported pulse height by a vWCPC indicates the fraction of the particle population generating an acceptably high pulse."**

Line 243-244: add reference to the figure when describing the result

**Response: we revised to add Fig 7(a) in line 403.**

Line 252: please add ref to the fig in supplemental material

**Response: we added the reference to the fig in supplemental material in line 407 - 412.**

Fig.7 Y-axis label. Please change to "Counting efficiency" for consistency

**Response: Thank you very much for catching that. We revised Fig. 7.**

Fig.S5: figure legends could be improved, currently displayed weird

**Response: we revised all the figures in the supplemental doc.**

Section: 3.4. "Chemical composition" – the authors refer to the particle material? Also this section could be made more concise.

**Response: we revised it to "Effect of particle chemical composition on the vWCPC counting efficiency". We also revised this section to make it more concise.**

Line 267-260: add reference to fig when describing these results.

**Response: we added "as shown in Fig. 8" in line 445.**

Line 270: what kind of analysis?

**Response: we removed this sentence because the simplified condensation effects analysis in the supplemental section didn't fully explain the chemical effect.**

Line 268 and 271: how to understand that counting efficiency statements "similar" and "affected" Please review this subsection.

**Response: we revised the sentence in line 445: "we get the counting efficiencies close to 1 for PSL, humic acid and AS particles, as shown in Fig. 8.".**

Line 272: you mention sucrose and humic acid as particle material used. Please discuss why and what these two represent.

Response: we revised line 441-443: "We choose two types of 100 nm aerosol particles: water-insoluble particles, such as oleic acid, humic acid particles and PSL, and highly hydrophilic ammonium sulfate and sucrose particles."

Wouldn't you expect that at 100 nm particle size these all particle material would be at 100% counting efficiency at ~900-1000 hPa? Why this is not the case e.g. for sucrose?

Response: This observation is consistent with the previous study (Hering et al., 2017). One possible explanation is that the chemical similarity between the seed particle material and the working fluid also affects the detection efficiency of vWCPC (Wlasits et al., 2020).

Fig. 8. Please add information in the caption on the size of these particles and their material.

Response: we revised the caption to "Fig. 8, CPC 3789 counting efficiency at different operating pressure with four different types of 100 nm aerosol particles (PSL, humic acid, AS, and sucrose), when the temperature conditions are Tcond =  $27 \,^{\circ}$ C and Tini = 59 °C. "

Line 285-286: butanol-based CPC? Which model from which study? Please add a reference or additional information.

Response: we revised in line 490-492: "For the butanol-based CPC, such as CPC (TSI, 7610), the CPC cut-off size was strongly influenced by the temperature difference between the saturator and condenser (Hermann and Wiedensohler, 2001; Kangasluoma and Attoui, 2019)."

Line 287: please add ref to the figure.

**Response: we revised in line 492-493: "We observed that this trend held well for the vWCPC using AS particles, as shown in Fig. 9."**

Fig.9: In legend you state "30C TSI". Please specify for which particle material and pressure settings these results were obtained. Also why there is a difference in the slope between the results indicated as TSI and of this study? Do you have data points you could add for 30 C to see if your data agrees with the one referenced as TSI 30C?

Response: we revised the caption to "Fig. 9, CPC 3789 counting efficiency changes as a function of the different operating pressures (500, 700, and 910 hPa) using AS particles, when the initiator temperature is 59 °C, and the moderator temperature is 10 °C, at two different conditioner temperatures (a) Tcond = 24, (b) Tcond = 27 °C." We also added one more sentence about TSI 30 C in line 498-500. "Note that the counting efficiency curve from TSI at 30 °C was derived and fitted using the AS particle classified by a custom-made Vienna-type different mobility analyzer(Wlasits et al., 2020). " The authors may consider adding a summary table in supplemental material that presents various temperature and pressure settings investigated, and resulting cut off sizes for certain particle material used.

**Response: we revised section 3.5 to include particle material used in the cut-off size characterization. In legends of Fig 9 (a) and 9(b), we included the temperatures and pressures for the characterization.**

It would be interesting to see if vWCPC flow rate (and so the instrument response) is influenced by the changing pressure. This CPC uses an internal pump, is that correct? What was the flow rate at various pressure settings? The authors may want to add a plot in the supplemental material.

**Response: Yes, the CPC uses an internal pump and a critical orifice to control the 0.3 Ipm volumetric flow. We monitored the critical pressure ratio, which is derived by dividing the absolute pressure downstream of the orifice by the absolute pressure upstream of the orifice. This value was maintained below 0.528 for all the testing reported in this manuscript.**

Section 4. Conclusion: I suggest to revise this section. Currently some detailed information is missing and not all information that is given is clear. I would be interested to see what is the meaning of these results, and what's the outlook for operating the vWCPC on the aircraft? Do the authors have any recommendations? If yes, what would the optimal setting? Any suggestions for studying the performance of this CPC under 500 hPa? Is such study planned? Little is said about the advantage of the WCPC over other CPCs, and how these results compare to results from other studies. Any limitations or benefits over other studies could be mentioned too.

**Response: we revised the conclusion section.**

Line 301: "was modified to report environmental pressure"? Please review.

**Response: we revised to "inlet operating pressure" in line 519.**

**REFERENCES**:**

Williamson, C., Kupc, A., Wilson, J., Gesler, D. W., Reeves, J. M., Erdesz, F., McLaughlin, R., and Brock, C. A.: Fast time response measurements of particle size distributions in the 3–60 nm size range with the nucleation mode aerosol size spectrometer, Atmos. Meas. Tech., 11, 3491–3509, https://doi.org/10.5194/amt-11-3491-2018, 2018.

---

## Author Comment (AC2)

General comments

This paper reports some good work to characterize a commercial water-based CPC, designed for operation near sea level, at reduced pressures, and furthermore that such operation is possible with only minor modification of the instrument.  This capability is important for capturing the vertical profile of aerosol in airborne research, and water is a far safer alternative to the traditional butanol working fluid.  The measurements are backed up with a modelling study which also helps explain the CPC response, and limitations therein, to pressure changes.  I provisionally recommend this manuscript for publication pending changes described below.

The manuscript should have been more carefully proofread prior to submission.  Although a number of syntax and typo issues are noted below, all authors are strongly encouraged to have another go at careful proofreading as part of their response to reviews.

**Response: We sincerely appreciate the comments and suggestions from our reviewer.  Thank you very much for considering the publication of our manuscript. We address your specific comments below (also in blue). The line number corresponds to the change-tracked version.**

Specific comments

Why choose 500 mb as the lower pressure limit?  Research aircraft reach sub-100 mb and balloons even lower pressures.  The graphs may provide the answer, but some statement about why you didn't investigate lower pressure ought to be included, possibly in the introduction.  Especially if lower pressure was attempted unsuccessfully, include a few words about what was tried and the outcome.
**Response: we revised the instruction section in line 109-112. "For the pressure condition lower than 500 hPa, the current simulation was highly nonlinear and the returned solution fails to converge. In addition, we observed inconsistent behavior in one of three vWCPC we tested. Thus, this manuscript focused on the measurements and modeling that were done over the pressure range from 500 hPa to 1000 hPa."**

Abstract:  should give the pressure range investigated instead of 'low pressure conditions'

**Response: we revised in line 15-16 to add the pressure range.**

Line 29:   The statement about detecting sub-100 nm particles isn't strictly correct, since optical scatter detection is possible down to ~50-60 nm (UHSAS, DMT).

**Response: we revised the sentence to "sub-50 nm"**

Please make your use of 'vWCPC', 'wCPC', etc. consistent throughout.
**Response: we revised wCPC to vWCPC.**

Lines 120-121

Equalizing diffusion losses by matching flows also requires matched tube lengths. Please indicate whether this was the case in your setup. Also this apparently contradicts the previous section which said two different flows were used.

**Response: we revised line 146-147 to explain that we focused on 0.6 lpm aerosol inlet flow. In addition, we also revised line 173-174 "Both CPC 3789 and A.E. were run at 0.6 lpm inlet flow with matched tubing lengths to ensure equal diffusive particle loss in the aerosol pathway."**

Lines 137-138:   To make it clear you are describing other work here, suggest starting the 2nd sentence this way:  "They first computed the temperature and humidity profiles were using..."  The next sentence describing the Hering et al. configuration doesn't seem to add anything to this paper and could be omitted.

**Response: we revised section 2.3 accordingly.**

Lines 162-163:  What's the sample rate?  How many samples during each 5-min. run, approximately?  Otherwise we can't make sense of any standard deviations.

**Response: The sampling rate is 1 Hz (added to line 290). Thus, during each 5 min, we collected around 300 data points.**

Line 183:  The title of section 3.2 doesn't reflect the combination of model and observation within it.

**Response: we revised to "Simulation-aided pressure dependence study of the vWCPC counting efficiency at different operating temperatures"**

Line 198:  '...one 8 nm see particle grew to a smaller size...'  Unclear what is meant here - please rewrite.  Also change 'no matter' to 'whether'

**Response: Sorry for the misleading, it is "… one 8 nm seed particle grew to a smaller size".**

Lines 234-235:  It's not at all clear how the observations of 100 nm particles in Fig. 6 inform the behavior of particles at smaller sizes down to 15 nm.

**Response: we revised in line 384-385 and added Fig S3, "Although we presented data with 100 nm particles in Fig. 6, we observed a similar trend with particles down to 15 nm (Fig. S3)." Then, we also revised in line 390-392, "Combining the observations from Fig 6 and Fig S3 and the simulation in Fig. 4 and 5, when the simulated saturation is over 1.3, the counting efficiency maintained close to 1 for aerosol particles larger than 8 nm under low-pressure conditions (500 – 920 hPa)."**

Sect. 3.3:

Lines 241-242:  Does this mean it does not count pulses that are below some trigger threshold? Insert 'of maximum' after '90%'

**Response: we revised in line 400-402," The reported pulse height by a vWCPC indicates the fraction of the particle population generating an acceptably high pulse. The manufacture's manual described the pulse height being above 90% for moderate concentrations (~10-5,000 cm$^{-3}$)."**

Lines 245-246:  Suggest rewriting this sentence:  'Meanwhile, for 100 nm particles at 500 hPa, the threshold concentration for a 10% reduction in counting efficiency was about...'

**Response: we revised accordingly in line 404-405.**

Lines 249-251:  This is an important point that is not easy to see, and isn't shown in Fig. 7.  Suggest pointing to the supplement figure:  'Additionally, Fig. S5(b) shows there is no significant...'

**Response: we revised accordingly in line 409.**

Also:  change '...simulation estimated 10% reduction of s and Dp happened...' to 'simulated 10% reduction of s and Dp...'

**Response: we revised accordingly in line 411.**

Sect. 3.5 & Conclusions:

Should include a statement about the cut-off being less sharp than for the TSI standard settings, as the price for operation over a much wider pressure range.  This can be important in the presence of a large ultra-fine particle mode.
**Response: we added in line 498-505, "Note that the counting efficiency curve from TSI at 30 ℃ was derived and fitted using the AS particle classified by a custom-made Vienna-type different mobility analyzer under a standard operation condition (Wlasits et al., 2020).  As a result of operation over a much wider pressure range, the cut-off curve derived by this study is less sharp than for the TSI standard settings. "**
* * *
Technical corrections, by line or label:

53   comma after 'system'

**Response: added in line 82.**

61   strike 'her'

**Response: removed 'her' in line 90.**

67   comma after 'initiator'

**Response: added in line 112.**

68  change ')' after 2017 to a semicolon

**Response: changed in line 113.**

85   change 'guided' to 'guide'

**Response: changed in line 132.**

101  change 'positive pressure difference' to 'positive difference'

**Response: change in line 157.**

104-106

Suggest recasting this sentence to 'Thirdly, we added pressure transducers (Baratron 722B, MKS Instruments, Inc., Andover, MA, USA) to the vWCPC inlet and exhaust lines.'

**Response: changed in line 160-161.**

Fig. 1    Please label at least one of the filters in the figure, or identify in the caption.

**Response: we labeled the HEPA filter connected to the flow buffer.**

150  change '…which shows that the flow rate varied…' to 'which shows that when the flow rate increased…'

**Response: changed in line 274.**

158  change 'maintained 2~4' to 'maintained in the range 2-4'

**Response: changed in line 286.**

Fig. 3    Please state in the caption that the configuration is TSI's standard.

**Response: stated in line 313.**

188  Change both instances of 'was' to 'is'

**Response: changed in line 338.**

196  Strike the comma after 'both'

**Response: removed the comma in line 345.**

Fig. 4    It would be helpful to separate the bunched contour labels.

**Response: In the publication version, we can make the figure larger to separate the labels.**

Caption:  change 'temperature is 59' to 'temperature at 59'

**Response: changed in line 357.**

Fig. 5 caption:  change 'temperature is 59' to 'temperature at 59'

**Response: changed in line 364.**

227  change 'imitator' to 'initiator'

**Response: changed in line 382.**

236  change 'than the' to 'than when the'

**Response: change in line 393.**

240  insert 'efficiency' after 'counting'

**Response: inserted in line 399.**

Fig. 7 caption:  change ' temperatures were set' to ' temperatures set'

**Response: changed in line 437.**

268  change 'efficacies' to 'efficiencies'

**Response: change in line 445.**

285, 287  change 'was' to 'is'

**Response: changed in line 490.**

Fig. 9:  Drop the '25' tick label, both panels

**Response: we updated Fig. 9.**

307  change 'the above phenomena' to 'this behavior'

**Response: the sentence was revised in 525.**

308, 309  'at' is preferable to 'under' because 'under 1000 hPa' might be read as _less than_ 1000 hPa, e.g.

**Response: changed in line 526.**

Author contributions:  Gregory Lewis and Maynard Havlicek are omitted.
* * *
Supplement

After Eqn. 1 (for D_th):  the expression for D_th at other than STP is missing.
**Response: we assume that the pressure effect on the D_th is negligible.**

After Eqn. 2 (for D_va):  '(0.21 by Steve)'  what is this?

**Response: we have removed this additional note.**

Please give a source for the Antoine equation empirical constants.

**Response: added the reference.**

Please define all quantities in the diffusion time expressions.  Many are not.

**Response: The diffusion time expressions for simulating the simplified condensation effects were included in the previous publication by Lathem and Nenes (2011). We briefed the main equations in the supplement document and added the reference of the diffusion time expressions in line 78-80.**

The primes in Eqn. 6 (for Gamma) appear to be on the wrong characters, and their meaning is not identified.

**Response: Γ is a growth parameter that depends on the droplet size and the water vapor mass transfer coefficient.**

After Eqn. 3 (for Le):  '...as detailed in the supplement.'  This IS the supplement.  Do you mean  '...as detailed above'?

**Response: changed to "as detailed above".**

Fig. S2

Main title and y-axis labels need attention.  May just be PDF rendering problems, but in my copy the RH unit reads '(l)', and the title repeats 'Relative humidity (l)' three times.

Caption:  change 'calculated at the' to 'calculated along the'.

**Response:  we revised Fig. S2 and caption.**

After Eqn. 4 (for s):  'and assuming dT/dz=G' should just be 'G=dT/dz'

**Response: changed.**

Following paragraph:  change 'hence lower the droplet size' to 'hence a lower droplet size'

**Response: changed**

After Eqn. 7 (for C-dot):  change 'assume' to 'write'

**Response: changed**

After Eqn. 9 (for Dp/Dp0):  change 'set to equal' to 'set equal'

**Response: removed "to"**

Following sentence:  remove ', which' after 0.01

**Response: removed ","**

Next sentence: should be Fig. S3, not 3.

**Response: changed to Fig. S4.**

Fig. S3

X- and Y-axis labels, and the legend, need attention.  Subscripts and superscripts are displaced from other characters.  Again, this could just be a PDF rendering problem.

Please narrow the legend box to avoid clipping the traces if possible.

Are all the 'exit' subscripts necessary?  Maybe strike them and instead include 'at the initiator exit' somewhere in the caption?

Caption:  change 'setting temperatures' to 'temperature settings'

**Response: revised Fig. S4.**

2nd paragraph after Fig. S3, 1st line:  change 'function as' to 'are functions of'; and again Fig. S3, not 3.

**Response:  changed in line 127**

The accommodation coefficients are discussed here, but don't appear in any expressions.  How were they used?

**Response: we added additional information after equation (10).** "Where $k_a'$ is a modified thermal conductivity described as equation (17.72) by Seitnfeld and Pandis (2016). $D_{va,P}'$ is a modified mass diffusivity described as equation (17.62) by Seitnfeld and Pandis (2016). When calculated the above two modified conductivities, we assume the value of the thermal accommodation coefficient ($\alpha_T$) was set equal to the mass accommodation coefficient ($\alpha_c$) in this simplified analysis."

Fig. S4 caption

Change 'for the droplet size is 3 mm when the droplet exit the initiator' to 'for a droplet size of 3 um exiting the initiator' (making sure to correct mm to um).  Please add a sentence describing the insets.

**Response: we revised the caption.**

Figs. S5, S6

Axis labels and legend text are far too small.  Legend superscripts are dramatically separated from their base characters.

**Response: we revised both figures.**

Fig. S5(b)

There are no magenta points on the plot, but they are in the legend (24C, N=2e4, 20 nm case).

**Response: we revised Fig. S6(b). The previous figure was messed up during the pdf conversion.**

Fig. S5 caption, last sentence:  strike 'were' and the following 'with', and correct 'temrperature'.

Fig. S6 caption: change 'temperature is 59' to 'temperature at 59'.

**Response: we revised captions of Fig. S5 and S6.**

Table S1

Asterisk-bullet mismatch for the note; use the same symbol.
**Response: revised.**

References: Seinfeld and Pandis (2016), Lathem and Nenes (2011), Nenes and Seinfeld (2003) are not given anywhere, main or supplement.

**Response: we updated the reference for the supplement.**

---

## Author Response (AR2)

Thanks for submitting your manuscript to AMT. The manuscript is suitable for publication with technical revision. Please find below some technical comments that should be addressed in the final submission.

**Dear Editor,**

**We sincerely appreciate the comments and suggestions from you.  Thank you very much for considering the publication of our manuscript. We address your comments below (also in blue). The line number is corresponding to the change-tracked version.**

Throughout the manuscript, when using trade names, please be sure to include the manufacturer and their place of business. For example, you do this for Igor Pro, but not for COMSOL, or for TSI, or for FC-43, or for Lindberg/Blue.

**Response: we added the business place for COMSOL (line 255), TSI (line 15), FC-43 (line 79, the original paper didn't mention the business place for this chemical, thus, we included the reference), Lindberg/Blue (line 187).**

Line 197: Use "vWCPC 3789" for consistency.

**Response: revised in line 194.**

Line 258: Use "vWCPC 3789" for consistency.

**Response: revised in line 257. We revised the manuscript to use "vWCPC 3789"**

Line 264: Change "decrease" to "decreased" (the rest of the paragraph is in past tense).

**Response: changed to "decreased" in line 290.**

Line 303: Change "condenser" to "conditioner"

**Response: changed to "conditioner" in line 334.**

Fig. 4. Please replot using a color scale that is more readable by those with color-vision impairment. Within Igor Pro, EOSSpectral11 is a good choice.

**Response: we revised the color scale in Fig. 4. We used MATLAB for plotting and followed a color suggestion from a figure designer.**

Fig. 5. For the color-vision impaired, pleas change one of the two filled circle symbols to a different shape (e.g., open circle or triangle). Same for the asterisk symbols.

**Response: revised Fig. 5.**

Fig. 5. Should there be two sets of dashed lines for each of the panels, showing the starting and ending points of the moderator?

**Response: we added the additional dash lines in Fig. 5.**

Line 356: Change "initator" to "initiator".

**Response: changed to "initiator" in line 390.**

Fig. 6 caption: Change "CPC 3789" to "vWCPC 3789".

**Response: change in the caption.**

I'm a bit confused at lines 374-375. "The reported pulse height. . . indicates the fraction of the particle population. . . ." I don't believe that pulse height indicates a fraction. Please restate more clearly. Further, "pulse height is above 90%. . .". 90% of what? Its nominal height? Same on line 378.

**Response: Sorry for the confusion, we revised in line 414-416. "The reported "pulse height" by a vWCPC 3789 is not the pulse height value produced by the light scattering signal. It is a calculated parameter, which indicates the fraction of the particle population generating an acceptably high pulse. The manufacturer's manual states that the pulse height is above 0.9 for moderate concentrations (~10-5,000 cm$^{-3}$)."**

The first paragraph in section 3.3 mixes up verb tenses, with "is" being used when referring to past measurements (e.g., line 379). Please use past tense ("was") consistently throughout this manuscript when referring to previously collected observations. However, when referring to a current analysis or interpretation of the data (e.g., line 385), still use "is". The last sentence should be, "When the pulse height was less than. . . ."

**Response: we revised the section 3.3.**

Figure 7. In panel a, please explain the units on the y-axis. This looks like normalized pulse height (relative to pulse height at 1000 hPa).

**Response: see the above explanation for Fig. 7. We changed the y axis to "pulse height ratio".**

Line 425. What is meant by the "absorbing qualities" of humic acid? Are you saying that light absorption by humic acid has an effect? Or are you talking about absorption of water? Please specify what is meant.

**Response: we revised it to "…light absorbing properties of humic acid" in line 474.**

Fig. 8. Please use different symbol types for the different curves, and refer to the instrument as a "vWCPC 3789" in the caption.

**Response: we revised Fig. 8.**

Line 452. Pleas change to "For butanol-based CPCs such as the TSI 7610, the CPC. . . ."

**Response: changed in line 519.**

Line 456. Please change tense from "was" to "is".

**Response: changed in line 522.**

Line 446. Please change to "vWCPC 3789".

**Response: changed 512.**

Line 461. Change from "the AS particle" to "AS particles".

**Response: changed in line 527.**

Line 461. I don't understand why using a Vienna-style differential mobility analyzer affects the sharpness of the cut-point at surface pressure. If the DMA type is irrelevant, just say that that TSI response curve was determined at near-sea-level pressure; thus the curve is sharper.

**Response: we revised it to "Note that the counting efficiency curve from TSI at 30 ℃ was derived and fitted using AS particle classified by a custom-made different mobility analyzer (using an aerosol to sheath flowrate ratio of 1:100) under a near-sea-level pressure" in line 526-528.**

Fig. 9. Please use different symbols instead of just different colors. Change "CPC 3789" to "vWCPC3789".

**Response: we revised Fig. 9 and changed the caption.**

Line 486: Change in verb tense; change "is" to "was".

**Response: changed in line 554.**

Under "Competing interests", you may wish to state which authors have a commercial interest in the success of the vWCPC instrument.

**Response: we revised the "competing interests".**

Please thoroughly edit the references and ensure they meet Copernicus guidelines. Formatting is inconsistent. For example, sometimes article titles are capitalized; sometimes not. This is a result of reference manager-type software, which always messes up the formatting and needs to be checked by hand very thoroughly. It will save the Copernicus editorial staff considerable time if you go ahead and make these changes.

**Response: Thank you very much for mentioning the software shortcoming. We revised the reference.**

Supplemental material: I do not see a revised Supplemental Material file with tracked changes. Please provide one with tracked changes attached to the revised submission.

**Response: The upload website didn't include the option to upload a change-tracked file for supplemental material, only for the main manuscript. Thus, we didn't upload one. We will combine it with the change-tracked main manuscript.**

[revised manuscript text omitted]

Where $k_a'$ is a modified thermal conductivity described as equation (17.72) by Seitnfeld and Pandis (2016). $D_{va,P}'$ is a modified mass diffusivity described as equation (17.62) by Seitnfeld and Pandis (2016).

When calculated the above two modified conductivities, we assume the value of the thermal accommodation coefficient ($\alpha_T$) was set equal to the mass accommodation coefficient ($\alpha_c$) in this simplified analysis.

To further simplify the equation (8) and (9), more convenient forms can be derived if $\dot{C}$ is explicitly written as a function of $D_p$, N, and $\Gamma$. The average droplet size $\overline{D_p} = (1/N)\sum_n N_i D_{pi}$.

$$\dot{C} = \frac{\pi R^* T \rho_w}{2 M_w} \Gamma N \overline{D_p} S \quad\quad (11)$$

If we write $\Phi = \frac{\pi^2 R^2 R_g R^* T^3 \rho_w}{\Delta H_v G Q P_s M_w}$, equation (8) can be simplified as

$$\frac{S}{S_0} = \frac{1}{1 + \frac{\Phi}{2}\Gamma N \overline{D_p}} \quad\quad (12)$$

Where $R^*$, $M_w$, $\rho_w$ are the universal gas constant (8.314 J/mol/K), the molecular weight and density of liquid water.

The simplification of the droplet size depression equation results from equation (9) and (10)

$$\frac{D_p}{D_{p0}} = \left(1 + \Phi \Gamma N \overline{D_p}\right)^{-1/2} \quad\quad (13)$$

The value of $\alpha_c$ was varied from 1 for rapidly activating aerosol to 0.01 which for slowly activating aerosol. However, based on the estimation, this variation did not significantly affect the saturation and droplet size, as shown in Fig. S4. Additionally, reducing the conditioner temperature also has influenced (<20% with the 15% reduction of s) the saturation profile. Previous studies showed that the droplet size exiting the moderator tube might have up to 90% particle loss if the droplet size is larger than 10 $\mu$m (Chen and Pui, 1995; Fletcher et al., 2009; Takegawa and Sakurai, 2011). Meanwhile, the signal-to-noise ratio is too high for small droplets. Thus, this simulation assumed the droplet size exiting the initiator is between 1 to 7 $\mu$m.

[Figure]

[Figure]

**Fig. S4. Predicted supersaturation depletion and droplet size depression ratio as a function of aerosol number concentration.**
**Results are shown for different mass accommodation coefficients and conditioner temperatures setting.**

The wCPC monitors the height of the pulses generated in the optical detector and reports a status
parameter to indicate the percentage of the sampled particles, which have an acceptably high pulse.
Although the exact droplet size detected by the detector is unknown, this pulse height parameter
indirectly shows insufficient particle growth in the detector chamber.

The saturation depletion and the droplet size depression are function of the aerosol number concentration at the ambient condition (1 atm), as shown in Fig. S4. The 10% reduction of s and $D_p$ is
predicted for $N_{1\mu m} \sim 6 \times 10^4$ (cm$^{-3}$), the mean droplet size at the initiator's exit is 1 µm with the
conditioner temperature setting is 30 °C.  Under the same temperature setting, if the mean droplet size
at the exit of the initiator should be 7 µm to make sure the detector counts the particles, the 10%
reduction of s and $D_p$ happened when the $N_{7\mu m} \sim 8.5 \times 10^3$ (cm$^{-3}$). With the conditioner's temperature
decreased to 24 °C, the threshold concentration ($N_{1\mu m}$ and $N_{7\mu m}$) for the 10% reduction of s and $D_p$
increased about 15% ($N_{1\mu m} \sim 7 \times 10^4$ (cm$^{-3}$) and $N_{7\mu m} \sim 1 \times 10^4$ (cm$^{-3}$)) from the concentration values
under the 30 °C conditioner temperature. Thus, the droplet size at the initiator's exit determines the
aerosol number concentration limits due to the saturation depletion and the droplet size depression.

The simulation results shown in Fig. S4 suggest that the droplet size at the initiator's exit should be
larger than 3 µm under the low-pressure. We examined the effect of the operating pressure on the 10%
reduction threshold theoretically, as shown in Fig. S5. The theoretical analysis suggests that the 10%
reduction threshold ($N_{3\mu m}$) is about $1.94 \times 10^4$ (cm$^{-3}$) at 1 atm, when the conditioner temperature is
24 °C. Based on the theoretical analysis, with the decrease of the operating pressure, the 10% reduction
threshold of $N_{3\mu m}$ reduced about 5% of the aerosol concentration ($1.85 \times 10^4$ (cm$^{-3}$) at 0.5 atm).

[Figure]

[Figure]

**Fig. S5. Predicted supersaturation depletion and droplet size depression ratio as a function of aerosol number concentration.**
**Results are shown for a droplet size of 3 μm exiting the initiator and the conditioner temperature is 24 °C. The insets are**
**zoomed in plots for the narrowed S/S₀ range.**

[Figure]

(a)

[Figure]

                                              (b)

Fig. S6. The water depletion due to the aerosol number concentration, illustrated by (a) the pulse height generated in the
optical detector, (b) the counting efficiency as a function of the inlet pressure. Results are shown with the conditioner
temperatures set at 24 °C and 30 °C, with the initiator temperature is 59 ℃ and the moderater temperature is 10 ℃.

[Figure]

Fig. S7. Predicted droplet size evolution along the growth tube of the CPC 3789 under the different conditioner temperatures (30 ℃, 27 ℃, and 24 ℃), with the initiator temperature at 59 ℃ and the moderater temrperature is 10 ℃. Starting particle size is 20 nm.

| Deleted: is |
| --- |
| **Formatted:** Font: 9 pt |

**Table S1.** Properties of tested aerosol particles.

| Properties | Ammonium sulfate | PSL | Sucrose | Humic acid | Oleic acid | Water |
|---|---|---|---|---|---|---|
| Molecular weight (g/mol) | 132.14 | N/A | 342.3 | 227.17 | 282.47 | 18.02 |
| Melting point | 235 ℃ | 100-110 ℃* | 186 ℃ | 300 ℃ | 13.4 ℃ | 0 ℃ |
| Density (g/cm³) | 1.77 | 1.055 (20 ℃) | 1.59 | 1.77 | 0.895 | 0.997 (20 ℃)) |
| Water solubility | 70.6 g/100 g water | insoluble | greater than or equal to 100 mg/mL at 66° F | insoluble | insoluble | N/A |
| Reference | https://en.wikipedia.org/wiki/Ammonium_sulfate | https://www.thermofisher.com | https://pubchem.ncbi.nlm.nih.gov/compound/Sucrose | https://pubchem.ncbi.nlm.nih.gov/compound/90472028 | https://www.britannica.com/science/oleic-acid | https://en.wikipedia.org/wiki/Water |

*Glass transition temperature